# Postglacial fire history and interactions with vegetation and climate in southwestern Yunnan Province of China

Xiayun Xiao[1], Simon G. Haberle[2], Ji Shen[1], Bin Xue[1], Mark Burrows[2], and Sumin Wang[1]

[1]State Key Laboratory of Lake Science and Environment, Nanjing Institute of Geography and Limnology, Chinese Academy of Sciences, Nanjing 210008, China
[2]Department of Archaeology and Natural History, College of Asia and the Pacific, Australian National University, Canberra, Australian Capital Territory 0200, Australia

*Correspondence to*: Xiaoyun Xiao (xyxiao@niglas.ac.cn); Ji Shen (jishen@niglas.ac.cn)

**Abstract.** A high-resolution, continuous 18.5 ka (1 ka=1000 cal yr BP) macroscopic charcoal record from Qinghai Lake in southwestern Yunnan Province, China reveals postglacial fire frequency and variability history. The results show that three periods with high-frequency and high-severity fires occurred during the periods 18.5-15.0 ka, 13.0-11.5 ka, and 4.3-0.8 ka, respectively. This record was compared with major pollen taxa and pollen diversity indices from the same core, and tentatively related with the regional climate proxy records with the aim to separate climate- from human-induced fire activity, and discuss vegetation-fire-climate interactions. The results suggest that fire was mainly controlled by climate before 4.3 ka and by the combined actions of climate and humans after 4.3 ka. Before 4.3 ka, high fire activity corresponded to cold and dry climatic conditions, while warm and humid climatic conditions brought infrequent and weak fires. Fire was an important disturbance factor and played an important role in forest dynamics around the study area. Vegetation responses to fire after 4.3 ka are not consistent with those before 4.3 ka, suggesting that human influence on vegetation and fire regimes may have become more prevalent after 4.3 ka. The comparisons between fire activity and vegetation reveal that evergreen oaks are flammable plants and fire-tolerant taxa. *Alnus* is a fire-adapted taxon and a nonflammable plant, but density of *Alnus* forest is a key factor to decide its fire resistant ability. The forests dominated by *Lithocarpus/Castanopsis* and/or tropical trees and shrubs are not easy to ignite, but *Lithocarpus/Castanopsis* and tropical trees and shrubs are fire-sensitive taxa. Fire appears to be unfavourable to plant diversity in the study area.

## 1 Introduction

Fire is a natural, recurring episodic event in almost all types of vegetation and is one of the primary natural disturbance factors in forest ecosystems (Harrison et al., 2010). It has a strong influence on the extent and diversity of forest resources, carbon cycles and global climate change. Major forest fires often cause serious harm and huge losses to the forest, environment and human livelihoods. In order to decrease the harm and losses, it seems necessary to prevent the occurrence of fires and suppress fires. However, some ecologists have discovered that fire suppression (or the exclusion of fire-catalyzed practices) can significantly degrade biodiversity (Pyne, 1998). On the other hand, some

studies suggest plant diversity increases after fires (Trabaud and Lepart, 1980; Stocker, 1981; Nasi et al., 2002). Thus, there is still debate about the effects of fire upon biodiversity, which may reflect different effects of fire on regional biodiversity as well as different scales of fire intensity and frequency that can occur. In order to make a reasonable strategy for forest fire management for a region, we need to better understand effects of fire upon biodiversity in this

region. At the same time, a better understanding of the driving mechanisms of forest fires is also imperative. Knowledge of past fire activity is a key to understanding the present day and for making sustainable management policies for forest ecosystems (Ali et al., 2009). Thus, it is necessary to reconstruct past long-term fire frequency histories, which can inform the current conceptual models of forest recovery after fire and provide guidance for forest management strategies in areas affected by frequent fires (Tinner et al., 1999; McWethy et al., 2013; Morales-Molino

et al., 2013; Kloster et al., 2015). Fire histories have been reconstructed by using the time series of fire atlases (Rollins et al., 2001; Le Page et al., 2008), collection and analysis of fire-scarred trees (Arno and Sneck, 1977; Bake and Dugan, 2013), and charcoal particle analysis in peat and lake sediments (Whitlock and Larsen, 2001; Holz et al., 2012; de Porras et al., 2014). However, sources of fire atlases and fire-scarred trees are limited, and their time spans are relatively short (Brunelle and Whitlock, 2003), whereas the charcoal records from peat and lake sediments can provide

long continuous fire frequency history and allow vegetation-fire-climate interactions to be examined (Gavin et al., 2007; Morales-Molino et al., 2013). Recently, black carbon was also gradually used as evidence of fire history reconstruction (Schmidt and Noack, 2000; Wang et al., 2013; Wolf et al., 2014) and its subtypes of char and soot have potential to differentiate between smoldering and flaming fires (Han et al., 2012, 2016).

At present, studies on fire history and the interactions between long-term vegetation, fire, and climate are mainly

concentrated in North and South America, parts of Europe and Oceania (https://www.paleofire.org/index.php?p=CDA/index&gcd_menu=CDA). Based on these studies, methods and numerical approaches to reconstruct long term fire histories have been actively advanced. China has only started relatively late in this field. Most studies on fire history are concentrated in northern China (Li et al., 2005; Huang et al., 2006; Jiang et al., 2008; Han et al., 2012; Li et al., 2013; Wang et al., 2013; Zhang et al., 2015a), while studies in

southern China are relatively few (Sun et al., 2000; Luo et al., 2001; Gu et al., 2008; Xiao et al., 2013; Zhang et al., 2015b). In southwestern China, to the best of our knowledge, only Gu et al. (2008) reconstructed fire history for the past 2.0 ka using microcharcoal and Zhang et al. (2015b) reconstructed fire history for the past 18.5 ka using black carbon. These studies on fire history in China have mostly concentrated on the relationship between fire activity and climatic change on orbital to suborbital timescales (Sun et al., 2000; Luo et al., 2001; Li et al., 2005; Huang et al., 2006;

Jiang et al., 2008; Zhang et al., 2015a, b). Studies of past fire frequency and magnitude and its relationship to climate, human activity and vegetation dynamics are however, uncommon for China (Gu et al., 2008; Wang et al., 2013).

The climate in Yunnan Province is characterized by distinct wet and dry seasons with warm-wet conditions in

summer and mild-dry conditions in winter, which are typically conducive to frequent forest fires (Xu, 1991). Consequently, Yunnan Province is a zone of high fire frequency in China. The study of forest fire events for the period 1982-2008 in Yunnan Province shows that the rate of forest loss due to forest fire in different vegetation zones are variable (Chen et al., 2012). The highest rate of forest loss occurs in semi-humid evergreen broadleaved forest dominated by *Cyclobalanopsis glaucoides* and evergreen oaks and *Pinus yunnanensis* forest. The rate of forest loss in monsoon evergreen broadleaved forest dominated by *Castanopsis* and *Lithocarpus* is relatively low. The rate of forest loss in tropical rainforest and monsoon forest zones is however, lower than that in the former two vegetation types (Chen et al., 2012). A low resolution charcoal record from Qinghai Lake in southwestern Yunnan Province of China was briefly mentioned in Xiao et al. (2015). Here, we presented a high-resolution and continuous macroscopic charcoal record spanning about 18.5 ka from the same core, and specifically analyzed the fire event history and fire regime dynamics through time in southwestern Yunnan Province based on the charcoal record. Combined with the climate records and vegetation histories since 18.5 ka, reconstructed in Xiao et al. (2015), the main controlling factors of forest fire dynamics and the relationship between fire activity and vegetation were examined in this study. These results provide new evidences for studies on fire frequency variability and vegetation–fire–climate interactions in southwestern China, which is important for improving our predictions of the influence of climatic change on future fire activity and planning appropriate management policies for forest ecosystems.

## 2 Regional setting

Qinghai Lake (25°7'56.75" N,98°34'19.16" E, 1885 m a.s.l.) is located in the northeastern Tengchong County of southwestern Yunnan Province, southwestern China (Fig. 1). Southwestern China is characterized by the alternation of high mountain ranges running roughly north to south (Qionglai Mountain, Daxue Mountain, Shaluli Mountain from east to west) and parallel, deep and narrowly incised river valleys (such as Dadu River, Yalong River, Jinsha River). The large span of longitude and latitude, and altitudinal differences (ranging from about 1000 to above 5000 m a.s.l.) in southwestern China result in diverse vegetation types and various vertical vegetation belts in the region (Xiao et al., 2011). Gaoligong is the westernmost mountain among the mountains over 3000 m a.s.l. in southwestern China (Fig. 1). Qinghai Lake is situated to the west of the Gaoligong Mountain, and the topography of its west and south is below 2500 m a.s.l. Qinghai Lake is a volcanic dammed lake with a catchment area of 1.5 km$^2$ and a surface area of c. 0.25 km$^2$ (in 1990) (Wang et al., 2002). The average and maximum water depths were 5.2 m and 8.1 m in 2010, respectively. Currently fed by precipitation, groundwater and surface runoff, the lake is without a natural outlet.

The study region is characterized by a subtropical humid monsoon climate. It is mainly affected by the warm-humid airflow from the Indian Ocean and Bengal Bay in summer and by the southern branch of the Westerlies

in winter. Because of its southwestern location in Yunnan Province, the climate in the region is warm and very humid in summer and mild and moderately dry in winter. Tengchong meteorological station near Qinghai Lake records a mean annual temperature of 15.4°C and mean annual precipitation of 1506 mm (Fig. 1c). Most of the precipitation is concentrated in the rainy season from May to October, which is 85% of the annual precipitation (Xiao et al., 2015).

Qinghai Lake is located within a zone of semi-humid evergreen broadleaved forest in Hengduan Mountains of western Yunnan Province, whose south is adjacent to a zone of monsoon evergreen broadleaved forest of southwestern Yunnan Province (Wu et al., 1987). Due to the strong influence of human activities, the catchment of Qinghai Lake is mainly covered by plantation forests such as *Taiwania cryptomerioides*, *Cunninghamia lanceolata*, and *Alnus nepalensis*. The vertical vegetation belts on the west slope of the southern Gaoligong Mountain varies gradually from a semi-humid evergreen broadleaved forest (<2200 m a.s.l.) to a mid-montane humid evergreen broadleaved forest (2200-2800 m a.s.l.), *Rhododendron* shrubland (2800-3000 m a.s.l.), a sub-alpine shrub meadow (3000-3500 m a.s.l.), and sparse vegetation in rock debris (>3500 m a.s.l.) from bottom to top (Qin et al., 1992).

In our published pollen record from Qinghai Lake (Xiao et al., 2015), *Alnus*, evergreen oaks and *Lithocarpus/Castanopsis* are dominant tree taxa. The proportion of tropical trees and shrubs comprised mainly of *Altingia*, *Ficus*, *Pentaphylax*, *Schima*, *Symplocos*, *Camellia*, *Eurya*, and *Mallotus/Macaranga* is relatively high in the forests. And *Pinus*, *Betula* and deciduous oaks also often occur. At present, studies on combustibility of these plant species are mainly concentrated in experimental analysis. The trade standard for forestry in People's Republic of China (State Forestry Administration of China, 2008) suggests that *Pinus* spp. and *Quercus* spp. are identified as a flammable class; *Alnus* spp., *Castanopsis* spp. and *Schima* spp. are identified as a nonflammable class; and *Betula* spp., *Cryptomeria* spp. and *Cunninghamia lanceolata* fall in between these two classes. The experimental results about the combustion characteristics of 48 tree species from south China show that the fire-resistant ability of *Alnus nepalensis*, *Schima superba*, *Camellia oleifera*, *Altingia gracilipes*, *Exbucklandia populnea*, and *Myria rubra* is very strong (Tian et al., 2001). The combustion test results of 40 common tree species from Jiangxi Province suggest that *Symplocos setchuensis*, *Schima superba*, *Castanopsis sclerophylla*, and *Camellia oleifera* are the strongest fire-resistant tree species (Zhen et al., 2012). *Ficus microcarpa*, *Eurya macartneyi*, and *Pentaphylax euryoides* are selected as tree species for a biological fire prevention belt in regional cities because of their strong fire resistence (Lu, 2005). Combustion experiments of 25 woody plants from Kunming, Yunnan Province show that *Alnus nepalensis*, *Camellia aleifera*, and *Eurya groffii* are considered to be nonflammable tree species (Li et al., 2009).

## 3 Material and methods

### 3.1 Sediment sampling, laboratory analysis and dating

A 832 cm long sediment core (TCQH1) was extracted in waters 6.3 m deep near the centre of Qinghai Lake, in November 2010 using a UWITEC piston corer. The core was sectioned at 1 cm intervals. Samples were stored at 4°C until analyzed. Macroscopic charcoal particles (>125 μm in diameter) were extracted from 1 $cm^3$ samples at contiguous 1-cm intervals. Charcoal samples were soaked in 20 ml of 5% sodium hexametaphosphate for >24 hours and 20 ml of 10% $H_2O_2$ solution for 24 hour to disaggregate the sediment (Huerta et al., 2009). Samples were gently washed through a 125 μm mesh sieve and the residue was transferred into gridded petri dishes and counted under a stereomicroscope at magnifications of 40×. The age-depth model for the core has been established in the previously published study (Xiao et al., 2015), based on the $^{210}$Pb and $^{137}$Cs dates and the AMS $^{14}$C dates and using the recently developed Bayesian method (Blaauw and Christen, 2011). According to the model, the bottom date is presumed to be 18.5 ka, and the average temporal sampling resolution is ~22 years for the macroscopic charcoal record. The sedimentation rate before ~0.9 ka is relatively low and steady, with a mean of 0.0385 cm/a. Between ~0.9 ka and 1952 AD, the rate is evidently higher at 0.124 cm/a. After 1952 AD, the rate is far higher than the earlier periods, with a mean of 0.741 cm/a (Fig. 2).

### 3.2 Selection of percentages or concentrations of the main pollen taxa

Pollen concentration is usually used as a proxy indicator of vegetation density or biomass productivity and thus climatic condition (Grosjean et al., 2007). However, apart from biomass productivity, some other factors such as the lithology, sedimentation rate, input of inwashed material, detritus content and within-lake sedimentary processes may confuse the records of real changes in pollen concentration (Hicks and Hyvärinen, 1999). In this study, biomass productivity and climatic conditions revealed by total pollen concentration are not exactly consistent with those disclosed by pollen percentage assemblage (Fig. SF1). For example, the lowest pollen concentrations during the period 14.5-13.0 ka BP might not indicate unfavourable climatic conditions, but rather high detritus content under conditions of rising humidity and intensified surface run-off. In addition, the tendencies of the concentrations and percentages for the six main pollen taxa are almost consistent except for some differences during the period 14.5-13.0 ka BP (Fig. SF2). Thus, considering that the impact factors of the pollen concentration are complicated in this study, we use only pollen percentages for the main pollen taxa.

### 3.3 Pollen diversity indices and standardized method

Richness and evenness are two important components of biodiversity. Different diversity indices reflect species richness as well as evenness but with different weight (Odgaard, 1999). The richness index only counts the total number of taxonomic units, whereas the number of individuals is not considered (Magurran, 1988), thus it is easily

influenced by rare species. Simpson's index is primarily influenced by the relative frequency or representation of individuals (namely species evenness) (Simpson, 1949), which depends mainly on dominant species and is less sensitive to species richness (Odgaard, 1999). Analytic arguments, mathematical models, and simulations indicate that relationship between richness and evenness is simple, positive, and strong (Stirling and Wilsey, 2001). However, other studies indicate that richness and evenness may be independent measures in diversity (Smith and Wilson, 1996; Wilsey and Potvin, 2000), and their relationships may vary with ecological effects. In order to evaluate the plant diversity and explore the relationship between richness and evenness around the study area, palynological richness index and Simpson's reciprocal index were adopted in this study.

Palynological richness index is the number of different pollen types in every pollen sample. Because pollen counts in samples may differ, rarefaction analysis was used to estimate palynological richness (Birks and Line, 1992), which was computed using the vegan package for R (http://CRAN.R-project.org/package=vegan) (Oksanen et al., 2016). Simpson's reciprocal index (1/D) was calculated as

$$\frac{1}{D} = \frac{N(N-1)}{\sum_{i=1}^{S} n_i(n_i-1)}$$

where $n_i$ is the number of individuals in the $i$ th pollen types, $N$ the total number of individuals and S the total number of pollen types (Xiao et al., 2008). The higher values of this index indicate the greater sample evenness.

In this study, we use min-max normalization to standardize pollen percentages, which can eliminate the influence of their values and better visualize the interrelation among percentage variations of the selected pollen types in the same panel of one figure. The formula is $x^* = \frac{x_i - \min}{\max - \min}$. Where $x^*$ is the standardized data of the selected pollen type; $x_i$ is percentage of this pollen type in the $i$ th sample; max is the maximum value of this pollen type percentage; min is the minimum value of this pollen type percentage.

### 3.4 Fire event identification

Fire events were recognized by separating the macroscopic charcoal accumulation rates (CHAR; particles/cm$^2$/a) into CHAR background (BCHAR) and CHAR peak (PCHAR) components by using CharAnalysis software (Higuera et al., 2008; http://charanalysis.googlepages.com/). BCHAR was determined with a 500-year Lowess Smoother, robust to outliers, and it is the slowly varying trend in charcoal accumulation which may represent gradual changes in regional fire activity and/or charcoal production per fire. PCHAR was taken as residual after subtracting BCHAR from CHAR, representing local fire episodes (namely one or more fire events occurring in the duration of a peak). The threshold

value for charcoal peak detection was set at the 95[th] percentile of a Gaussian mixture modeling noise in the CHAR peak time series (Higuera et al., 2008). Fire frequency (episodes/ka) is the sum of the total number of fires within a 1000-a period, smoothed with a Lowess Smoother. Peak magnitude (particles/cm$^2$/peak) is the total charcoal influx in a CHAR peak and varies with fire size, severity, proximity, and taphonomic processes (Whitlock et al., 2006; Huerta et al., 2009; Walsh et al., 2010). The magnitude of CHAR peaks can be used as a proxy for fire severity (Ali et al., 2012), indicating further degrees of organic matter loss or burned biomass.

## 4 Results

The macroscopic charcoal and fire activity records were divided into six zones (from TCCC-1 to TCCC-6) (Fig. 2) based on their visual inspection and referring to palynological zonation boundaries, in order to facilitate the comparison with major pollen taxa and previous vegetation reconstruction for the same core (Xiao et al., 2015). The characteristics of the charcoal record, fire event reconstruction and pollen diversity indices in these zones were described as follows. In order to make the link with the published paper based on pollen data (Xiao et al., 2015), the pollen zonation (from TCQH-1 to TCQH-7) was kept in Figure 2.

**Zone TCCC-1** (18.5-15.0 ka), corresponding to the pollen subzones TCQH-1a and TCQH-1b: Charcoal concentration and CHAR (including BCHAR) in this zone were relatively high. Charcoal concentration averaged 585.6 particles/cm$^3$. CHAR ranged from 0 to 72.4 particles/cm$^2$/a with an average of 16.9 particles/cm$^2$/a. 19 fire episodes were registered by charcoal peaks. Peak magnitude varied greatly from 32.6 to 2189.2 particles/cm$^2$/peak with an average of 446.8 particles/cm$^2$/peak. Fire frequency ranged from 2.4 to 7.4 episodes/ka with an average of 4.9 episodes/ka. Palynological richness index was very low (mean of 53) and Simpson's reciprocal index was relatively low (mean 9.1). The vegetation type during the period was semi-humid evergreen broadleaved forest dominated by evergreen oaks with relatively more dry-tolerant herbs such as *Artemisia* and Chenopodiaceae (Xiao et al., 2015).

**Z**one **TCCC-2** (15.0-13.0 ka), corresponding to the pollen subzones TCQH-1c and TCQH-2a: Charcoal concentration and CHAR decreased markedly, averaging 114.5 (ranging from 6.7 to 412.3) particles/cm$^3$ and 5.2 (ranging from 0.1 to 21.4) particles/cm$^2$/a, respectively. Only one significant fire episode was noted in the bottom of this zone (peak magnitude: 58.4 particles/cm$^2$/ peak) and two fire episodes were registered in the top of this zone (peak magnitudes are both at 63 particles/cm$^2$/episode). Fire frequency declined first from 5.7 episodes/ka to 0.2 episodes/ka and then increased to 5.1 episodes/ka with an average of 2.3 episodes/ka. Palynological richness index increased markedly, averaging 59. Simpson's reciprocal index increased first to 12.2, then decreased to an average of 8.2. In this zone, the vegetation type was still semi-humid evergreen broadleaved forest, but the dominant plant species evergreen oaks retreated first (in the pollen subzone TCQH-1c) and then expanded gradually (in the pollen subzone TCQH-2a), whereas *Alnus* expanded first (in the pollen subzone TCQH-1c) and then decreased (in the pollen subzone TCQH-2a),

and dry-tolerant herbs (e.g. *Artemisia*) decreased evidently (Xiao et al., 2015).

**Zone TCCC-3** (13.0-11.5 ka), corresponding to the pollen subzone TCQH-2b: This zone had the highest charcoal concentration and CHAR for the entire core, averaging 636.2 (ranging from 92.3 to 1806.2) particles/cm$^3$ and 25.1 (ranging from 0.4 to 67.7) particles/cm$^2$/a, respectively. Fire episodes occurred frequently, and the magnitudes of CHAR peaks average 426.4 particles/cm$^2$/peak, ranging from 0.02 to 1984.4 particles/cm$^2$/peak. Fire frequency was the highest for the profile, average 9.2 episodes/ka. In this zone, palynological richness and Simpson's reciprocal indices were relatively low, averaging 57 and 7.5, respectively. The vegetation type during the period was semi-humid evergreen broadleaved forest dominated by evergreen oaks with low *Artemisia* (Xiao et al., 2015).

**Zone TCCC-4** (11.5-4.3 ka), corresponding to the pollen zones TCQH-3, TCQH-4, and TCQH-5: Charcoal concentration and CHAR remained very low in this zone. Average charcoal concentration was 113.8 particles/cm$^3$. CHAR values ranged from 0 to 31.1 particles/cm$^2$/a with an average of 4.1 particles/cm$^2$/a. Three fire episodes were registered in the lower part of this zone (at 11.3, 10.4 and 10.2 ka, respectively) with low peak magnitudes (253.5, 32.7 and 29.3 particles/cm$^2$/peak, respectively). In addition, three fire episodes were registered at ca. 6.7, 5.8 and 5.7 ka with peak magnitudes of 18.3, 212.7 and 417.1 particles/cm$^2$/peak, respectively. Fire frequency was very low with an average of 1.1 episodes/ka. Palynological richness and Simpson's reciprocal indices were relatively high as a whole (averaging 70 and 11.6, respectively), and their highest values occurred during the period 10.4-8.5 ka. The vegetation type during the period changed gradually from semi-humid evergreen broadleaved forest to monsoon evergreen broadleaved forest dominated by *Lithocarpus/Castanopsis*, evergreen oaks and tropical trees and shrubs (Xiao et al., 2015).

**Zone TCCC-5** (4.3-0.8 ka) (corresponding to nearly the pollen zone TCQH-6) was characterized by a marked increase in charcoal concentration and CHAR compared with Zone TCCC-4, averaging 589.7 particles/cm$^3$ and 23.9 (ranging from 0 to 122.2) particles/cm$^2$/a, respectively. Ten fire episodes were noted in the zone. Peak magnitude varied from 26.3 to 939.4 particles/cm$^2$/peak (mean of 303.4 particles/cm$^2$/peak). Fire frequency averaged 2.6 episodes/ka, ranging from 0.7 to 4.8 episodes/ka. Palynological richness index had no obvious changes (mean of 68) compared with the previous zone, while Simpson's reciprocal index decreased rapidly (mean of 9.1). In this zone, the primary evergreen broadleaved forest was rapidly replaced by deciduous broadleaved forest dominated by *Alnus* (Xiao et al., 2015).

**Zone TCCC-6** (after 0.8 ka), corresponding to nearly the pollen zone TCQH-7: Charcoal concentration was very low, averaging 62.3 particles/cm$^3$. CHAR (including BCHAR) first decreased markedly to mean of 6.8 (ranging from 0.7 to 27.1) particles/cm$^2$/a, then increased to mean of 58.8 particles/cm$^2$/a since 1940 AD. Two fire episodes were registered at 0.3 ka and 1986 AD (peak magnitudes: 33.7 and 359.7 particles/cm$^2$/peak), respectively. Fire frequency was relatively low, averaging 1.8 (ranging from 0.7 to 3.3) episodes/ka. Palynological richness index declined

markedly (mean of 62), whereas Simpson's reciprocal index increased (mean of 10.9). In this zone, the greater open vegetation was present in the region, the area of *Alnus* forest quickly decreased, and the proportion of tropical trees and shrubs declined, whereas the secondary semi-humid evergreen broadleaved forest recovered (Xiao et al., 2015).

Palynological richness and Simpson's reciprocal indices indicate that relationship of plant richness and evenness is roughly positive before 4.3 ka, except that the increase of richness at 10.4 ka is slightly later than the increase of evenness at 11.5 ka. However, plant richness and evenness show an approximately inverse correlation after 4.3 ka.

## 5 Discussion

### 5.1 Fire history from Qinghai Lake

The results of fire event reconstruction based on the macroscopic charcoal record from Qinghai Lake reveals that fire frequency and peak magnitude were not constant over the last 18.5 ka in southwestern Yunnan Province. There are three periods with frequent and intensive fire episodes, occurring during 18.5-15.0 ka, 13.0-11.5 ka, and 4.3-0.8 ka. Between 15.0 and 13.0 ka, fire episodes first rapidly decreased in frequency and severity, then gradually increased. From 11.5 to 4.3 ka, fire frequency and severity was at a low level as a whole, and there were only six fire episodes with low severity at 11.3, 10.4, 10.2, 6.7, 5.8 and 5.7 ka. Between 0.8 ka and 1940 AD, only one weak fire episode was noted, and fire frequency and severity were relatively low. Since 1940 AD, due to the high sedimentation rate (Fig.2), CHAR (including BCHAR) increased markedly although the charcoal concentration was still very low, which indicates relatively severe fire activity.

The black carbon content of the same core reveals three periods of high fire activity: 18.5-15.0 ka BP, 13.0-11.5 ka BP, and 8.0-~0.9 ka BP (Zhang et al., 2015b). Comparison of fire activity results revealed by the two proxies shows that the start and end times of the first two periods of high fire activity are consistent, whereas the start times of the last period of high fire activity are different. The pollen record reveals that vegetation type was monsoon evergreen broadleaved forest dominated by *Castanopsis/Lithocarpus*, and the climate was warm and humid during the period 8.5-4.3 ka BP (Xiao et al., 2015). Carbonized plant remains can be distinguished when macroscopic charcoal particles are counted under the stereomicroscope, whereas they cannot be separated when black carbon is measured. Differential rates of decomposition and plant carbonization may occur during periods of high plant productivity, for example during warm and humid climate periods, and this may result in the different fire activity results during the period ~8.0-4.3 ka BP.

### 5.2 Regional climate changes in the southwest monsoon region

From the pollen record in Qinghai Lake, standardized data of *Lithocarpus/Castanopsis* (taxa indicating warm and

humid conditions), evergreen oak (indicating relatively temperate conditions), and herb (indicating relatively dry conditions) pollen percentages are selected as a synthetic climate proxy (Fig. 3d, Xiao et al., 2015). At the same time, the DCA (detrended correspondence analysis) axis 1 sample scores of pollen data from Qinghai Lake may also act as a climate proxy indicating climate changes from relatively cool and moderately dry conditions to warm and humid conditions from high scores to low scores (Fig. 3e, Xiao et al., 2015). The recent studies (Xiao et al., 2014a, b) suggest that the PCA (principal components analysis) axis 1 sample scores of the pollen data from Tiancai Lake in northwestern Yunnan Province can be mainly interpreted as temperature change from warm to cold conditions from low scores to high scores, respectively (Fig. 3f). Selection of the DCA or PCA in different studies is based on the underlying linearity of the data. Namely, the PCA is selected when most of the underling responses are linear or at least monotonic to the underlying latent variables, otherwise the DCA is selected (Xiao et al., 2014a). From the pollen record of Lugu Lake located on the boundary between Yunnan and Sichuan Provinces in southwest China, standardized data of *Tsuga* (indicating humid conditions), *Betula* (indicating relatively temperate conditions), and herb pollen percentages are selected as a synthetic climatic proxy (Fig. 3g, Wang, 2012). These pollen proxies from Qinghai Lake (Fig. 3d, e), Tiancai Lake (Fig. 3f), and Lugu Lake (Fig. 3g) reveal regional climatic changes since 18.5 ka in western Yunnan Province that included two cold events (the Heinrich Event 1 (H1) and the Younger Dryas cold event (YD)) and a warm period (the Bølling/Allerød warm period, B/A) during the last deglaciation. Because the different responses of the three study sites located in different latitudes and altitudes to global climate change and age uncertainty, the H1, B/A, and YD are considered to have occurred during the periods $17.5\pm0.5\sim15.4\pm0.4$ ka, $14.4\pm0.2\sim12.9\pm0.1$ ka, and $12.9\pm0.1\sim11.5$ ka in western Yunnan Province, respectively.

The time of the H1 is almost consistent with the date defined by Sanchez-Goñi and Harrison (2010, HS1 is dated between 18 to 15.6 ka). The B/A and YD are in good agreement in timing and duration with the B/A between 14.6 and 12.8 ka given by Grootes and Stuiver (1999) and the YD between approximately 12.8 and 11.5 ka given by Muscheler et al. (2008). The Holocene was clearly identified to begin at 11.5 ka. From 10.4 ka, the temperature and precipitation increased rapidly. The Holocene climatic optimum (HCO) occurred during the period $8.5\pm0.3\sim3.8\pm0.5$ ka. After $3.8\pm0.5$ ka the climate became cooler and drier, which may be accompanied by signals of human activity such as slash and burn (Xiao et al., 2015). This climatic trend is also recorded in other data from regions affected by the southwest monsoon, such as the $\delta^{18}O$ records from core NIOP905 in the western Arabian Sea (Fig. 3h) (Huguet et al., 2006) and core KL126 in the Bay of Bengal (Kudrass et al., 2001), and the sediment colour record from northeastern Arabian Sea (Deplazes et al., 2013).

**5.3 Climate forcing on fire occurrences**

Comparing the fire activity record from Qinghai Lake with the regional climate records in western Yunnan Province and the southwestern monsoon region (Fig. 3), it appears that the frequent and intensive fire activity during the periods 18.5-15.0 ka and 13.0-11.5 ka corresponded to the relatively cold period before the H1, during the H1, and during the YD. The low fire activity between 15.0 and 13.0 ka corresponded to a period of slight warming (between H1 and B/A) and the B/A warm period. The rare and weak fire activity between 11.5 and 4.3 ka corresponded to the gradual increases in the temperature and precipitation from 11.5 to 8.5 ka and the Holocene climatic optimum between 8.5 and 4.3 ka. Under the background of weak fire activity as a whole during the period 11.5-4.3 ka, six fire episodes at 11.3, 10.4, 10.2, 6.7, 5.8 and 5.7 ka may indicate six significant dry or highly seasonal rainfall periods. However, the abrupt cold event at 8.2 ka recorded in other palaeoclimate records from the southwest monsoon region (Kudrass et al., 2001; Dykoski et al., 2005) was not marked in the charcoal record. One reason may be that the relatively low temperature was not favourable to fuel desiccation. However, the more important reason may be that the climate was not too dry or rainfall seasonality was relatively low in the study area, because drought is the key factor influencing fire occurrences in southwestern China (Gu et al., 2008).

Paleofire studies at global scales reveal that high fire activity occurred during warm interstadials or interglacials, and low fire activity occurred during cold stadials or glacials (Power et al., 2008; Daniau et al., 2010; Mooney et al., 2011). These studies consider that the overall reduction in biomass was a severe constraint on fire regimes during the glacial. Namely, cold and dry climatic conditions influence plant biomass, therefore available fuel for fire is limited. As the study area is located in the southern subtropics, warm and humid climate conditions lead to diverse vegetation types, which means that climate change in a certain range may only result in changes in proportion of dominant trees or vegetation types, and may not cause obvious changes in plant biomass. Thus, the fuel biomass shall not be a constraining factor for fire activity in this study area, whereas the most important control factor of fire occurrence (or fire frequency) in the region is dry climate or high rainfall seasonality, particularly in the period prior to 4.3 ka, which is consistent with Gu et al. (2008)'s conclusion. In addition, some other paleofire studies in the monsoon region of China also suggest the same factor controlling fire activity as this study (Fig. 1a). The charcoal and pollen records from peat section at Qindeli, northeastern China show that the frequency and severity of fire are high during the dry period and low during the wet period (Li et al., 2005). A well-dated peat profile (i.e. the HE profile) with high resolution charcoal and pollen records from the Sanjiang Plain in northeastern China indicates that low frequency regional and local fires responded to the strong summer monsoon during the interval ~6.0–4.5 ka, and then the fire frequencies increased evidently with the decline of the summer monsoon (Zhang et al., 2015a). The pollen and charcoal records from a Jinchuan peat (northeastern China) indicate that a natural origin for fire event 1 (5.1 ka) was probably facilitated by drying environmental conditions and fire event 2 (1.3 ka) was caused by clearing (Jiang et al., 2008). The charcoal records in the Holocene loess–soil sequences (XJN: Xujianian, ETC: Ertangcun, JYC: Jiangyangcun, and

DXF: Dongxiafeng) over the southern Loess Plateau of China suggest that local wildfires occurred frequently during the late last glacial period and the early Holocene before 8.5 ka. During the Holocene climatic optimum between 8.5 ka and 3.1 ka, natural wildfires were largely reduced. Levels of biomass burning were very high during the late Holocene, when the climate became drier and historical land-use became more intensive (Huang et al., 2006). The charcoal and pollen records from the two deep sea cores 17940 (taken by Sino-German joint cruise "SONNE-95" in 1994) (Sun et al., 2000) and 1144 (taken by ODP Leg 184 in 1999) (Luo et al., 2001) in the northern part of the South China Sea disclose that the high strength and frequency of natural fire corresponded to the drier climate during glacials or stadials. This relationship between climate and fire in the monsoon region of China is different from that at a global scale (Power et al., 2008; Daniau et al., 2010; Mooney et al., 2011), which may reveal that regional differences in climatic conditions determine vegetation type, fuel biomass and fire weather. However, it provides regional evidence for understanding the relationship between climate and fire that may be different under different climate conditions.

## 5.4 Possible influence of human activity on fire activity

Archaeological records from Yunnan Province show that Neolithic culture began to emerge at around 4 ka or a little earlier (Han, 1981; Li, 2004). The pollen assemblage from Qinghai Lake shows the obvious increases in Poaceae and *Artemisia* pollen percentages and the rapid degradation of the primary evergreen broadleaved forest from 4.3 ka (Xiao et al., 2015). Poaceae grains with diameters greater than 40 μm were usually classified as cereal-type (Lamb et al., 2003). In the study, Poaceae grains (>40 μm) began to occur at 18.5 ka and its pollen percentages had no obvious change (Fig. SF3), indicating that Poaceae grains (>40 μm) didn't reflect changes of human activity and the criterion of cereal pollen (Poaceae grains with diameters greater than 40 μm) may not be applicable in this study area. Studies on the characteristics of modern Poaceae pollen in Yunnan Province are needed for redefining the criterion of cereal pollen for this study area. Because cereal pollen was not separated from Poaceae pollen, we cannot rule out the possibility of increase in human activity when Poaceae pollen percentages increased. Thus, the obvious increase in Poaceae pollen percentages, combined with the rapid degradation of the primary evergreen broadleaved forest and the archaeological records, reveal that human activity may have influenced vegetation changes from 4.3 ka in the study area. This raises the possibility that superimposed influence of human activities such as forest clearance and agricultural cultivation and climatic cooling and drying may have influenced the frequent and strong fire activity between 4.3 and 0.8 ka.

Historical documents show that the first administrative centre (Ruanhua Fu) was established in the early stage of Dali Kingdom (1.01–0.7 ka) in Tengchong (Editorial Board of Tengchong County Annals, 1995). Subsequently, population immigration and implement of the state collective farming system accelerated deforestation and reclaimed farmland, which prompted intensification of agriculture (Editorial Board of Tengchong County Annals, 1995). The

pollen record from Qinghai Lake suggests greater open vegetation presented in the region after 1.0 ka (Xiao et al., 2015). Due to the decrease of forest area caused by human activity, the capacity for soil and water conservation declined, resulting in significant soil erosion. The coeval rapid increase in sediment accumulation rate may be caused by loss of forest cover and increased soil erosion, which diluted the charcoal input and resulted in low charcoal concentration. The same low CHAR revealed the relatively low fire frequency and severity between 0.8 ka and 1940 AD, which may be caused by human suppression of fire in an intensively managed agricultural and urbanized environment. The relatively intensive and frequent fire activity since 1940 AD corresponds to burning and plundering events during World War II and subsequently rapid economic development at the expense of vegetation.

### 5.5 The relationship between fire activity and vegetation

Climate exerts the dominant control on the spatial distribution of the major vegetation types on a global scale. In the study area, when the climate is warmer and more humid than today, the proportion of hygrophilous and thermophilous components in the forest increases, and even vegetation shifts gradually to a monsoon evergreen broad-leaved forest or tropical rainforest or monsoon forest. When the climate is cooler and relatively drier than the present day, the proportion of tolerant-dry components in the forest increases, and even vegetation changes gradually into a mid-montane humid evergreen broadleaved forest (Wu et al., 1987). In addition to the dominant control of climate on vegetation, fire was also one of important disturbance factors in vegetation changes. Past effects of fire on vegetation may be revealed by correlations between charcoal and pollen records (Tinner et al., 1999, 2005; Colombaroli et al., 2007). In this study, vegetation types, pollen diversity indices and four main arboreal pollen types (tropical trees and shrubs, *Lithocarpus/Castanopsis*, evergreen oaks and *Alnus*) with the highest percentages in the pollen assemblage from Qinghai Lake were selected to discuss vegetation responses to fire by comparing them with the charcoal record (Fig. 2).

Among the three periods with high fire activity, the two periods with high fire activity occurred in semi-humid evergreen broadleaved forest, and the other one occurred in deciduous broadleaved forest dominated by *Alnus*. Whereas, the unusual or weak fire activity during the period 11.5-4.3 ka arose in vegetation shifting from semi-humid evergreen broadleaved forest to monsoon evergreen broadleaved forest (during the period 11.5-8.5 ka) and monsoon evergreen broadleaved forest (during 8.5-4.3 ka). These are consistent with the modern results on the highest rates of forest loss due to forest fire in semi-humid evergreen broadleaved forest and relatively low rates of forest loss in monsoon evergreen broadleaved forest (Chen et al., 2012).

Frequent and severe fire activity during the periods 18.5-15.0 ka and 13.0-11.5 ka was linked to high pollen percentages of evergreen oaks, indicating that evergreen oaks are flammable plants, which is consistent with *Quercus* spp. identified as flammable class in the trade standard for forestry in People's Republic of China (State Forestry

Administration of China, 2008). When the high fire activity began at 18.5 and 13.0 ka, evergreen oak pollen percentages declined gradually; whereas when the high fire activity ended at 15.0 and 11.5 ka, evergreen oak pollen percentages continued to decline for a short time after going through frequent and strong fire events, indicating that evergreen oaks are able to withstand a degree of burning and continue growing despite gradual damage from fires.

Thus, evergreen oaks are relatively fire-tolerant taxa although they are flammable plants, and fire activity does not lead to their considerable decreases. However, there are some different opinions about the responses of evergreen oaks to fire events. Trabaud (1990) suggests that evergreen oak can survive fire and can even re-sprout vigorously afterwards. Conedera and Tinner (2000) show that *Quercus ilex* is highly sensitive to fire disturbance under certain circumstances based on ecological studies. Reyes and Casal (2006) consider that fire has little negative effect on evergreen oak

germination. Colombaroli et al. (2007) suggest that evergreen oak declines synchronously with fire-sensitive *Abies*. The reason for different responses of evergreen oak to fire events is likely different environment and evolutionary pressures for evergreen oak.

In this trade standard (State Forestry Administration of China, 2008), *Alnus* spp. is identified as a nonflammable class, and the results of combustion experiments of *Alnus nepalensis* also suggest that it is a nonflammable tree species

(Tian et al., 2001; Li et al., 2009). Observations after an intensive fire on 17 April, 1995 in Anning County of Yunnan Province found that a patch of *Alnus nepalensis* forest was not burned during the fire, which is further support for the fire resistant nature of *Alnus nepalensis*-dominated forest (Liu et al., 1996). However, low-density *Alnus nepalensis* forest was completely burned in this fire suggesting that density of *Alnus* forest is a key factor in determining its fire resistant capacity (Liu et al., 1996), even if the results of combustion experiments suggest that it is a nonflammable

tree species. In our study, high *Alnus* pollen percentages correspond to high fire activity during the period 4.3-0.8 ka, which may be because *Alnus* forest around Qinghai Lake during the period 4.3-0.8 ka was relatively low density or it was not pure *Alnus* forest. This deduction is verified by high representation of *Alnus* pollen, namely modern pollen rain results in southwestern China show that *Alnus* pollen percentages are more than 70% in a patch of *Alnus* forest (Xiao, unpublished). The abundance of *Alnus* promptly increased after the endings of frequent and strong fire activity before

4.3 ka, indicating that *Alnus* is a fire-adapted taxon. The adaptability of *Alnus* to survive fire events has been detected in other regions such as Mediterranean area (Lago di Massaciuccoli, Tuscany, Italy) and the Oregon Coast Range of USA (Colombaroli et al., 2007; Long et al., 2007). The responses of *Alnus* to fire before 4.3 ka and after 4.3 ka are different, which may be influenced by human activity. *Alnus*, as a pioneer plant, responded rapidly to human activity from 4.3 ka (at the beginning of the last high fire phase) and increased prior to the end of the frequent and intensive

fire period (at 0.8 ka). The marked reduction of *Alnus* at ~1.0 ka is probably caused by selective clearance of the forest by people (Xiao et al., 2015).

The very low fire frequency and severity between 11.5 and 4.3 ka corresponded to the gradual increase in

*Lithocarpus/Castanopsis* and tropical arboreal pollen percentages, and high *Lithocarpus/Castanopsis* and tropical arboreal pollen percentages. When the frequent and intensive fire activity began at 4.3 ka, *Lithocarpus/Castanopsis* and tropical arboreal pollen percentages declined rapidly. These suggest that the forests dominated by *Lithocarpus/Castanopsis* and/or tropical trees and shrubs are not easy to ignite, which is in good agreement with the results of combustion experiments suggesting that *Castanopsis* spp., *Schima superba*, *Camellia oleifera*, *Altingia gracilipes*, *Symplocos setchuensis*, *Ficus microcarpa*, *Eurya macartneyi*, and *Pentaphylax euryoides* are nonflammable tree species. However, frequent and intensive fire resulted in rapid decreases in abundance of *Lithocarpus/Castanopsis* and tropical trees and shrubs. Referring to Tinner et al.' (2000) definition of fire-sensitive taxa, we consider that *Lithocarpus/Castanopsis* and tropical trees and shrubs are fire-sensitive taxa in the study area.

Our results show that high fire-episode frequency occurs in conjunction with forests comprised primarily of fire-adapted taxa and lower fire-episode frequency is associated with forest dominated by fire-sensitive taxa, which is consistent with the Oregon Coast Range, USA, study (Long et al., 2007).

## 5.6 Dynamics of plant diversity

A comparison of fire activity and pollen diversity indices suggests that post-fire reactions of plant diversity before 4.3 ka and after 4.3 ka are different. Before 4.3 ka, the high fire activity corresponded to relatively low plant diversity (including two aspects of richness and evenness), whereas the very low fire activity corresponded to relatively high plant diversity. After high fire activity ended at 15.0 ka, plant richness and evenness increased. The reason may be that some fire-sensitive species and some thermophilous elements responding to the obvious increase of temperature at this time occurred, resulting in the increase of richness. When high fire activity ended at 11.5 ka, plant evenness increased and plant richness had no obvious change. The possible reason why plant richness did not increase at this time is that the number of new types of fire-sensitive species and the number of disappearing types of fire-adapted species may be roughly equal, while a slight increase in temperature at this time is not sufficient to promote the emergence of species adapted to warmer temperatures during the late glacial transition. All these indicate that fire is not favourable to plant diversity, and may degrade plant diversity in the study area. A similar response of plant diversity to fires is documented in Southern France where floristic richness increased markedly after fire (Trabaud and Lepart, 1980). However, the other studies suggest that there is a significant loss of plant diversity after fires in southern Switzerland (Tinner et al., 1999) and northern Tuscany of Italy (Colombaroli et al., 2007). The reason for this apparent contradiction is that forest ecosystems are naturally fire-sensitive, because fire tends to destroy the large biomass investments of trees. In contrast, fire-adapted communities that might have originated partly as a consequence of forest disruption may be favoured or even preserved by fire incidence (Colombaroli et al., 2007).

For the entire profile, the highest plant richness and evenness synchronously occurred during the period 10.4-8.5

ka when the temperature and precipitation increased evidently. During this period different plant species (including trees and herbs) were competing for habitat space and there were no obvious dominant species. Plant richness and evenness declined slightly during the climatic optimum from 8.5 to 4.3 ka. During this period, the forest was dominated by a few warm and moist species such as *Lithocarpus/Castanopsis*. These indicate that an evident ameliorating climate was more favorable to plant diversity than a steady optimum climate.

After 4.3 ka, the high fire activity (during the period 4.3-0.8 ka) was linked to relatively high plant richness and relatively low plant evenness. The possible reason is that the fire-adapted pioneer plants (such as *Alnus*) responded promptly to disturbance of human activity from 4.3 ka and became the dominant species before the end of high fire activity. However, the intensity of human activity at this period, as well as the frequency and severity of fire, is not enough to promote disappearance of fire-sensitive or disturbance-sensitive plant species. The low fire activity after 0.8 ka corresponded to the decrease of vegetation richness and the increase of vegetation evenness, which may be caused by a stronger disturbance of human activity such as deforestation and selective clearance, resulting in some disturbance-sensitive plant species disappearing and proportion of a few dominant species declining. Overall, human activity is likely to be the main driver of altered fire regimes after 4.3 ka and the response of plant diversity (richness and evenness) to fire activity during this period.

**5.7 Implications for management strategy for forest fires**

Combustibility of different forest types and a fire-sensitivity ranking for the main plant types are helpful for forest management and restoration after fires. In the study area, semi-humid evergreen broadleaved forest dominated by evergreen oaks is flammable. In order to prevent potentially destructive fires in the future, there is a need to pay greater attention to the origins and control of fire events in these forests. Once semi-humid evergreen broadleaved forest dominated by evergreen oaks is burned out, this forest needs a longer time to regenerate after fires even if it is not influenced by human activity. *Alnus* is a nonflammable tree species and has the potential to be used as a fire suppressant. However, if *Alnus* is used to prevent fires, it is necessary to plant pure *Alnus* forest, because fire suppressant capacity of low-density *Alnus* forest or mixed-*Alnus* forest is relatively weak. In addition, *Alnus* forest will regenerate relatively quickly after fires if it is not impacted by further human activity. The forests dominated by *Lithocarpus/Castanopsis* and/or tropical trees and shrubs are not easy to ignite because of their relative low inflammability. However, if frequent and intensive fire events occur in or adjacent to the forests dominated by *Lithocarpus/Castanopsis* and/or tropical trees and shrubs there is a greater risk for these forests to be impacted by major fire events. Furthermore, the forests dominated by *Lithocarpus/Castanopsis* and/or tropical trees and shrubs do not regenerate well after fires highlighting the need to prevent or suppress fires in these highly vulnerable forest types.

In our study area, fire is generally not favorable to plant diversity, and if it is of high enough frequency and

intensity, fire may also deplete plant diversity. Thus, fire control and suppression may be one management strategy to maintain plant diversity in the study area, which will have the added benefit of reducing the risk of large forest fires and maintaining relatively high biodiversity in the future.

## 6 Conclusions

This study about linkages of fire history, climatic change, human activity, and vegetation demonstrates that fire was mainly controlled by climate before 4.3 ka and by the combined actions of climate and humans after 4.3 ka. The frequent and intensive fire activity occurred under cold and moderately dry conditions or during the periods accompanied by human activity. Low fire activity occurred under warm and humid conditions or during the periods when temperature and humidity are increasing.

Fire was an important disturbance factor in the vegetation changes and played an important role in the forest dynamics and characteristics of the flora around the study area. The comparisons between fire activity and vegetation reveal that evergreen oaks are flammable plants. The forests dominated by *Lithocarpus/Castanopsis* and/or tropical trees and shrubs are not easy to ignite. The combustibility of evergreen oaks, *Lithocarpus/Castanopsis* and tropical trees and shrubs is in good agreement with the results of combustion experiments of these plants. Although *Alnus* is a nonflammable tree species, high *Alnus* pollen percentages corresponded to high fire activity during the period 4.3-0.8 ka, which may be because the density of *Alnus* forest is a more important factor to decide its fire resistant ability. A fire-sensitivity ranking for main arboreal types in the study area shows that *Alnus* is a fire-adapted species; evergreen oaks are fire-tolerant taxa; *Lithocarpus/Castanopsis* and tropical trees and shrubs are fire-sensitive taxa. However, vegetation responses to fire after 4.3 ka are not consistent with that before 4.3 ka, which points to the possibility that vegetation and fire regimes were increasingly influenced by human activity after 4.3 ka. Before 4.3 ka, when vegetation was mainly influenced by climate dynamics, frequent and intensive fire activity reduced plant diversity, and plant diversity increased after fires. Fire control and suppression may be one management strategy to maintain plant diversity in the study area. Since vegetation was influenced by a mixture of natural processes and anthropogenic activity (after 4.3 ka), the post-fire responses of vegetation have changed and their relationships have become complex.

Our results are important for enhancing our predictions of the influence of climatic change on future fire activity and making appropriate management policies for forest ecosystems. However, these results came only from a single lake site, and further lacustrine sites from southwestern China remain necessary to improve our understanding of vegetation-climate-fire interactions.

*Acknowledgments.* This research was financially supported by the Program of Global Change and Mitigation (2016YFA0600501), the National Natural Science Foundation of China (41272188, 41572149 and 41372184). We thank Prof. Houyuan Lu and Xiangdong Yang

for giving us some constructive advice, and Dr. Shuchun Yao and Yong Wang for doing fieldwork. The paper has been strongly improved thanks to the comments of Editor Nathalie Combourieu-Nebout and four anonymous reviewers.

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

# Figures

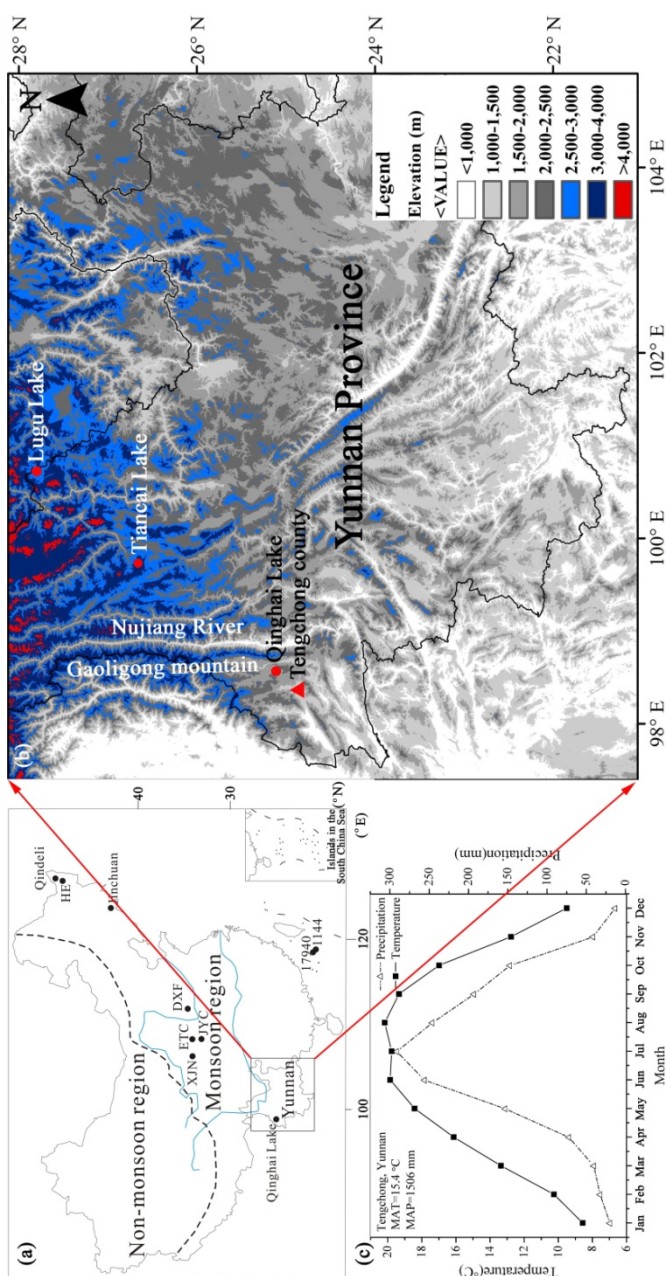

**Figure 1.** (a) The sketch map of monsoon and non-monsoon regions in China and other sites mentioned in text (XJN: Xujianian; ETC: Ertangcun; JYC: Jiangyangcun; DXF: Dongxiafeng). (b) Topographic map of southwestern China and the location of Qinghai Lake. (c) Climate diagram from Tengchong meteorological station near Qinghai Lake showing monthly temperature and precipitation. These data are 34-year climate averages for the period 1980-2013. MAT: mean annual temperature; MAP: mean annual precipitation.

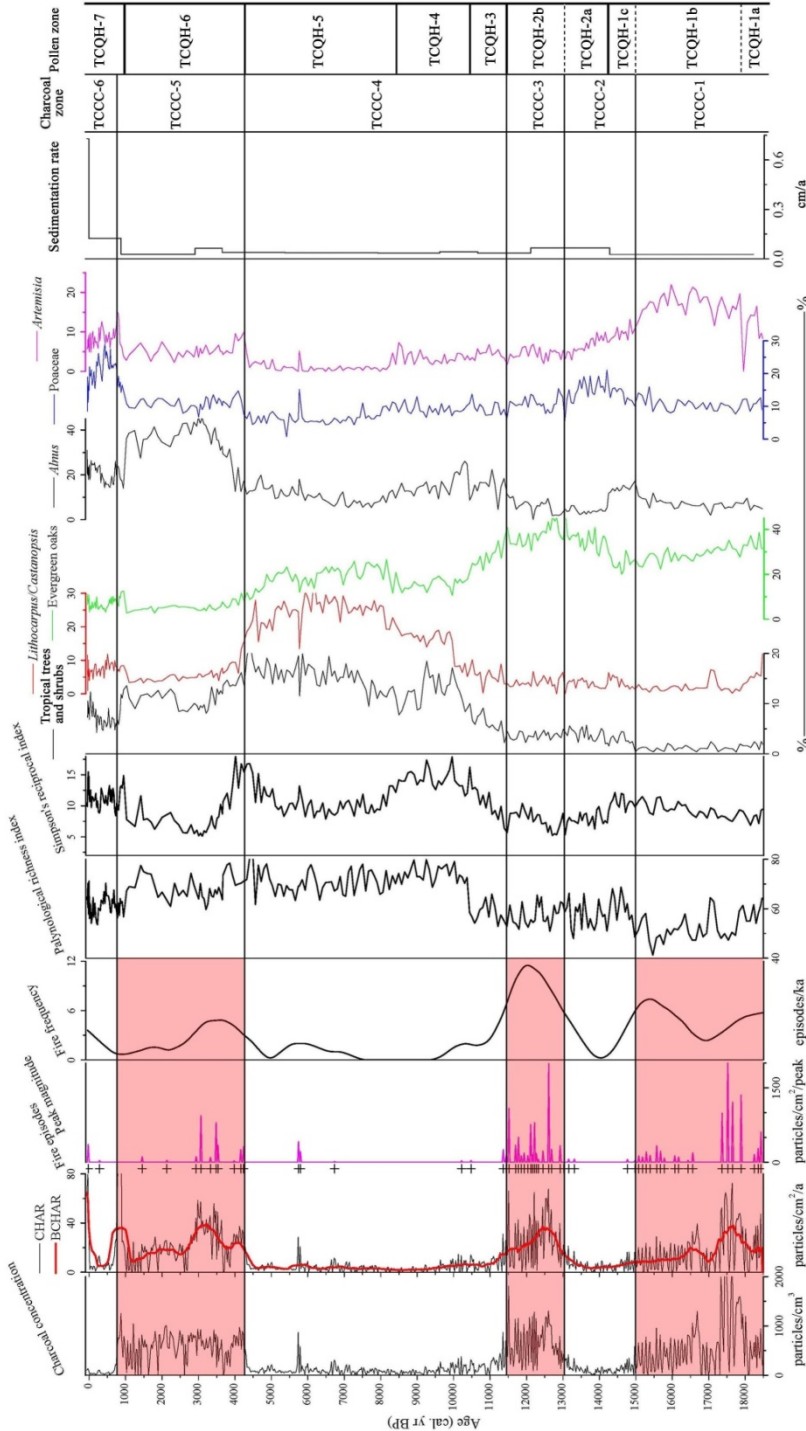

**Figure 2.** Results from charcoal analyses, pollen diversity indices, percentages of six main pollen taxa, and sedimentation rate for Qinghai Lake. The red shadings indicate the three periods with high fire activity.

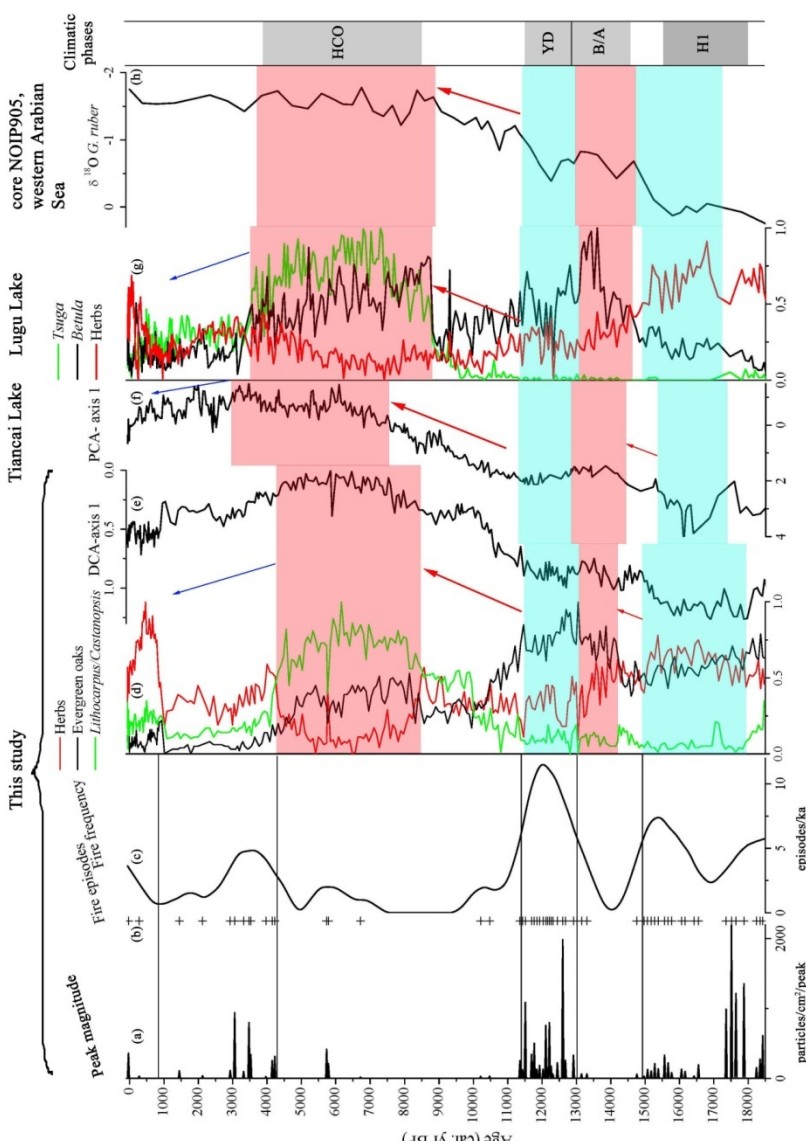

**Figure 3.** A comparison of fire activity from Qinghai Lake and climate proxies from the southwest monsoon regions. (a) Peak magnitude. (b) Fire episodes. (c) Fire frequency. (d) Standardized data of pollen percentages of *Castonopsis/Lithocarpinus* (green line), evergreen oaks (black line), and herbs (red line) from Qinghai Lake (this study). (e) DCA axis 1 sample score of pollen data from Qinghai Lake (Xiao et al., 2015). (f) PCA axis 1 sample score of pollen data from Tiancai Lake (Xiao et al., 2014a, b). (g) Standardized data of pollen percentages of *Tsuga* (green line), *Betula* (black line), and herbs (red line) from Lugu Lake (Wang et al., 2012). (h) The $\delta^{18}O$ record of the planktic foraminifera *G. rubber* for core NOIP905, western Arabian Sea (Huguet et al., 2006). The light green shadings indicate relatively cold and dry periods corresponding to the H1 and YD. The red shadings indicate relatively warm and humid periods corresponding to the B/A and HCO. The red lines with arrows indicate increases in temperature and precipitation. The blue lines with arrows indicate climate cooling and drying.