# Peer review of "Postglacial fire history and interactions with vegetation and climate in southwestern Yunnan Province of China"

_Climate of the Past, 2016_

## Referee Comment (RC1) · S.A.G. Leroy (Referee) · 29 Jul 2016

General comments This manuscript reports an interesting investigation of fire history combining not only charcoal but also various palynological parameters (including diversity). The results are robust and show clearly data that are different from global syntheses. This highlights the need to work also at a regional scale and understand the intricacies of climate change and vegetation types at a regional level.

My main scientific comment is why not use concentrations in addition (or even instead) of percentages for the pollen taxa (page 10, lines 18 and foll.). It could be more informative. Page 11, Lines 19-21: is not it the reverse? when there are fires only fire-adapted taxa survive?

Technical comments Page 3, lines 8-10: add a call to reference Page 3, Line 12: I suppose you mean the Westerlies Page 4, line 10 (and elsewhere): is not Page 12, line 11: lack of inflammability, or uninflammability. Is this what you mean? Page 12, line 24: replace "correlations" by "comparisons" or "parallel". Avoid "correlation" that suggests statistics were applied. Page 16, line 5: add italics to Latin names Figure 3: use the same major and minor ticks for a and b

––––––––––––––––––––––––––––––

---

## Referee Comment (RC2) · Anonymous Referee #2 · 2 Aug 2016

General comments:

This paper reports a high resolution of macroscopic charcoal record from Qinhai in the monsoon region of China for the last 18500 yrs cal BP. A lower resolution charcoal record is already published in Xiao et al. 2015 JQS as well as the pollen analyses. In addition to the high resolution charcoal record, the novelty concerns fire and vegetation interaction using fire episodes and frequency indexes compared to vegetation diversity indexes obtained from pollen assemblages from the same core. The authors reach the conclusion that fire occurs during cold and dry climate periods characterized by evergreen oaks and Alnus, and that fire lead to decrease in abundance of Lithocarpus/Castanopsis and tropical arbors. The material and methods section (and related

figures) should be reduced as this is already published in details in Xiao et al. 2015 JQS. The discussion section needs to be restructured with a clear description of fire adapted and fire-sensitive taxa found in the region today in order to discuss fire impact on vegetation through time, and a discussion about how climate/monsoon drives the vegetation in the region should be presented before discussing the possible role of fire on the vegetation. Specific comments below should help to improve the manuscript in this way.

Specific comments:

- Title: it seems from the reading of the title that the charcoal and pollen records are new – remove "based on charcoal and pollen records"

- Line 12 page 1: compared with the pollen record – already published in Xiao et al 2015 JQS. Please modify to read "compared with major taxa and diversity indexes" or something similar.

- Line 10 page 2: the reference of Power et al. 2008 is wrong here

- Line 21 page 2: "resulting in forest fire occurring frequently" please add some words here or in regional setting about the kind of vegetation that is burning today and the different fire adapted taxa and fire-sensitive taxa found in the region. This would help to follow the discussion.

- Line 25 page 2: would be good also to discuss macrocharcoal results and Black Carbon of the same core published in Palaeogeography, Palaeoclimatology, Palaeoecology, Volume 435, 1 October 2015, Pages 86-94 by Zhang et al.

- Material and Methods - section 3.1: the Table 1 and age model (Figure 3) are already published in details in Xiao et al. 2015 and in Zhang et al. 2015. Table 1 and Figures (3a and 3b) should be removed and references should be clearly indicated in the Material section 3.1. If the authors want to republish the age model, they should update their record from IntCal09 to IntCal13. For macroscopic charcoal analysis: please indicate

Interactive
comment

that a low resolution record is already published in Xiao et al. 2015. Please explain in what the high resolution charcoal record will help to understand fire, vegetation and climate in this region.

- Line 20 page 4: why standardizing pollen percentages? Please justify your approach.

- Line 7 page 5: fire episode magnitude –discrepancy between units (particles/cm2) and the total charcoal influx?

- Results – Chronology: remove this paragraph because this is already published in details in Xiao et al. 2015 – or shortened it and put this in the Material and Methods section – but clearly it is not new results. Keep the description of the temporal sampling of charcoal however.

- Section 4.2 Charcoal record, fire events and palynological diversity indices: this section is difficult to follow because the numbers reported in the text is not readable on Figure 4. Change the scale of the units reported in Figure 4. In addition, the authors describe the charcoal variations following pollen subzones determined in Xiao et al. 2015 but for pollen subzone TCQH-2 they changed this approach and determined subzone based on fire activity? As this paper is a focus on fire activity would be better to describe what is happening using fire activity zones and not following pollen subzones. Zone TCQH-4: only one fire event was detected – did the authors try several smoothing window and check whether this event remains in case of changing the smoothing background?

- Line 22 page 7: "In the last 50 years, the charcoal concentration was still very low, and the relatively high CHAR may be at least partly due to the high sedimentation rate." Using the charcoal accumulation rate (CHAR) or in other word the charcoal influx instead of concentration is supposed to avoid "wrong" signal of fire in terms of concentration due to dilution. This sentence is unclear.

Discussion section: - Line 23 page 7: same comment as above about the sedimenta-

tion rate.

- Line 16 page 8: While frequent fires appear to occur during the YD, it is unclear during the H1 giving the dates used by the authors (see sanchez Goni and Harrison 2010, QSR – HS1 is between 18 to 15.6 kyr cal BP) and the choice of using pollen zone to describe charcoal trend. In this case, low fire frequency is recorded during HS1.

- From line 29 page 8: Artemisia pollen percentages are also high in TCQH-1b. Why Poaceae and Artemisia would be indicative of human activities from 4.3 ka? "Super-imposed influence of human activities and climatic cooling and drying" what are the proxies that indicate a cooling and drying then?

- Line 9 page 9: again unclear about dilution and charcoal influx. Add the sedimentation rate curve to one of the figures.

- Line 13 page 9: "Paleofire studies at global scales reveal that high fire activity occurred during warm interstadials or interglacials, and low fire activity occurred during cold stadials or glacials (Power et al., 2008; Mooney et al., 2011; Marlon et al., 2013)". The reference of Mooney et al. 2011 is for the Australasia, regional scale. Add for interstadials and stadials Daniau et al. 2010 QSR. Marlon et al 2013 is for the Holocene only.

- Line 7-8 page 10: same comment about references

- section fire activity and vegetation: this section would benefit of a clear description of what are the fire adapted and fire-sensitive taxa found in the region today. In addition, it would be good to discuss how climate/monsoon drives the vegetation in the region before discussing the possible role of fire on the vegetation composition.

- Figure 2 is already published in Xiao et al. 2015.

Technical details: - Line 3 page 9 and others: "Edirotial Board" modify to read "Editorial"

---

## Editor Comment (EC1) · N. Combourieu Nebout (Editor) · 2 Sep 2016

Dear authors,

We have now received two reviews of your paper. All two reviewers ask you many questions that need your full details responses and reviewer 2 make substantial comments you have to follow.

You have now to post your replies to all the comments on the discussion forum that explain how you want to modify your manuscript. Please also prepare a revised version of your paper accordingly. In the revised version, I would like to see your corrections in track change mode.

[Figure]

sincerely Yours

Nathalie Combourieu-Nebout

---

## Author Comment (AC1) · 4 Sep 2016

Response to professor S.A.G. Leroy:

General comments:

This manuscript reports an interesting investigation of fire history combining not only charcoal but also various palynological parameters (including diversity). The results are robust and show clearly data that are different from global syntheses. This highlights the need to work also at a regional scale and understand the intricacies of climate change and vegetation types at a regional level.

Response: We are very pleased to have received a review from Professor S.A.G. Leroy

and we are grateful for her positive feedback on the manuscript. We address each comment with explanation as follows.

My main scientific comment is:

1) Why not use concentrations in addition (or even instead) of percentages for the pollen taxa (page 10, lines 18 and foll.). It could be more informative.

Response: Yes, we usually use pollen concentration as a proxy indicator of vegetation density or biomass productivity and thus climatic condition. However, apart from biomass productivity, some other factors such as the lithology, the sedimentation rate, input of inwashed material, detritus content, within-lake sedimentary processes et al. may confuse the records of real changes in pollen concentration (Hicks and Hyvärinen, 1999). In this study, biomass productivity and climatic conditions revealed by total pollen concentration are not exactly consistent with those disclosed by pollen percentage assemblage (Fig. S1). For example, the lowest pollen concentrations during the period 14.5-13.0 ka BP might not indicate unfavourable climatic conditions, and might be caused by the high detritus content under conditions of rising humidity and intensified surface run-off. In addition, the tendencies of the concentrations and percentages for the 6 main pollen taxa are almost consistent except for some differences during the period 14.5-13.0 ka BP (Fig. S2). Thus, considering that the impact factors of the pollen concentration are complicated in this study, we use only pollen percentages for the pollen taxa. Hicks S, Hyvärinen H, 1999. Pollen influx values measured in different sedimentary environments and their palaeoecological implications. Grana 38: 228-242.

2) Page 11, Lines 19-21: is not it the reverse? when there are fires only fire-adapted taxa survive?

Response: It concerns the following sentence: high fire-episode frequency occurs in conjunction with forests comprised primarily of fire-adapted taxa and lower fire-episode frequency is associated with forest dominated by fire-sensitive taxa. Fire-adapted taxa

are defined as "plant species are able to withstand a degree of burning and continue growing despite gradual damage from fires". Fire-sensitive taxa are defined as "plant species whose abundance will decrease rapidly when they undergo frequent and intensive fire". In this study, evergreen oaks and Alnus are fire-adapted taxa, and are flammable plants. Lithocarpus/Castanopsis and tropical arbors are fire-sensitive taxa, but they are not easy to ignite. Thus, we draw the above conclusion. There is a difference between frequent and intensive fire events and fire events in terms of strength and frequency. Forests are primarily comprised of fire-adapted taxa, which can not be considered as only fire-adapted taxa survive. Thus, we can not say that when there are fires only fire-adapted taxa survive.

Technical comments:

1) Page 3, lines 8-10: add a call to reference

Response: A reference is added. Wang, Y. F., Zhu, Y. X., Pan, H. X., Yin, Y.: Environmental characteristics of an acid Qinghai Lake in Tengchong, Yunnan Province, Journal of Lake Sciences, 14(2), 117-124, 2002 (in Chinese).

2) Page 3, Line 12: I suppose you mean the Westerlies

Response: Yes, thanks, it is done.

3) Page 4, line 10 (and elsewhere): is not

Response: Done.

4) Page 12, line 11: lack of inflammability, or uninflammability. Is this what you mean?

Response: Yes, it is uninflammability. Thank you very much for your careful observation.

5) Page 12, line 24: replace "correlations" by "comparisons" or "parallel". Avoid "correlation" that suggests statistics were applied.

Response: Thank you very much for your suggestion. We replace "correlations" with "comparisons".

6) Page 16, line 5: add italics to Latin names

Response: Done, thanks.

7) Figure 3: use the same major and minor ticks for a and b.

Response: We delete Figure 3 in the origin manuscript according to Referee2' comments.

————————————————————

**Fig. 1.** A comparison of pollen percentages for 6 main pollen taxa and total pollen concentration

[Figure]

**Fig. 2.** A comparison of pollen percentages and concentrations for 6 main pollen taxa

---

## Author Comment (AC2) · 5 Sep 2016

**General comments:**

This manuscript reports an interesting investigation of fire history combining not only charcoal but also various palynological parameters (including diversity). The results are robust and show clearly data that are different from global syntheses. This highlights the need to work also at a regional scale and understand the intricacies of climate change and vegetation types at a regional level.

**Response:** We are very pleased to have received a review from Professor S.A.G. Leroy and we are grateful for her positive feedback on the manuscript. We address each comment with explanation as follows.

**My main scientific comment is:**

1) Why not use concentrations in addition (or even instead) of percentages for the pollen taxa (page 10, lines 18 and foll.). It could be more informative.

**Response:** Yes, we usually use pollen concentration as a proxy indicator of vegetation density or biomass productivity and thus climatic condition. However, apart from biomass productivity, some other factors such as the lithology, the sedimentation rate, input of inwashed material, detritus content, within-lake sedimentary processes et al. may confuse the records of real changes in pollen concentration (Hicks and Hyvärinen, 1999). In this study, biomass productivity and climatic conditions revealed by total pollen concentration are not exactly consistent with those disclosed by pollen percentage assemblage (Fig. S1). For example, the lowest pollen concentrations during the period 14.5-13.0 ka BP might not indicate unfavourable climatic conditions, and might be caused by the high detritus content under conditions of rising humidity and intensified surface run-off. In addition, the tendencies of the concentrations and percentages for the 6 main pollen taxa are almost consistent except for some differences during the period 14.5-13.0 ka BP (Fig. S2). Thus, considering that the impact factors of the pollen concentration are complicated in this study, we use only pollen percentages for the pollen taxa.

Hicks S, Hyvärinen H, 1999. Pollen influx values measured in different sedimentary environments and their palaeoecological implications. Grana 38: 228-242.

[Figure]

Figure S1. A comparison of pollen percentages for 6 main pollen taxa and total pollen concentration.

[Figure]

Figure S2. A comparison of pollen percentages and concentrations for 6 main pollen taxa.

2) Page 11, Lines 19-21: is not it the reverse? when there are fires only fire-adapted taxa survive?

**Response:** It concerns the following sentence: high fire-episode frequency occurs in conjunction with forests comprised primarily of fire-adapted taxa and lower fire-episode frequency is associated with forest dominated by fire-sensitive taxa. Fire-adapted taxa are defined as "plant species are able to withstand a degree of burning and continue growing despite gradual damage from fires". Fire-sensitive taxa are defined as "plant species whose abundance will decrease rapidly when they undergo frequent and intensive fire". In this study, evergreen oaks and *Alnus* are fire-adapted taxa, and are flammable plants. *Lithocarpus/Castanopsis* and tropical arbors are

fire-sensitive taxa, but they are not easy to ignite. Thus, we draw the above conclusion.

There is a difference between frequent and intensive fire events and fire events in terms of strength and frequency. Forests are primarily comprised of fire-adapted taxa, which can not be considered as only fire-adapted taxa survive. Thus, we can not say that when there are fires only fire-adapted taxa survive.

**Technical comments:**

1) Page 3, lines 8-10: add a call to reference

**Response:** A reference is added.

Wang, Y. F., Zhu, Y. X., Pan, H. X., Yin, Y.: Environmental characteristics of an acid Qinghai Lake in Tengchong, Yunnan Province, Journal of Lake Sciences, 14(2), 117-124, 2002 (in Chinese).

2) Page 3, Line 12: I suppose you mean the Westerlies

**Response:** Yes, thanks, it is done.

3) Page 4, line 10 (and elsewhere): is not

**Response:** Done.

4) Page 12, line 11: lack of inflammability, or uninflammability. Is this what you mean?

**Response:** Yes, it is uninflammability. Thank you very much for your careful observation.

5) Page 12, line 24: replace "correlations" by "comparisons" or "parallel". Avoid "correlation" that suggests statistics were applied.

**Response:** Thank you very much for your suggestion. We replace "correlations" with "comparisons".

6) Page 16, line 5: add italics to Latin names

**Response:** Done, thanks.

7) Figure 3: use the same major and minor ticks for a and b.

**Response:** We delete Figure 3 in the origin manuscript according to Referee#2' comments.

**Response to Anonymous Referee #2: Interactive comment on "Postglacial fire history and interactions with vegetation and climate in southwestern Yunnan Province of China based on charcoal and pollen records" by Xiayun Xiao et al.**

**General comments:**

This paper reports a high resolution of macroscopic charcoal record from Qinghai in the monsoon region of China for the last 18500 yrs cal BP. A lower resolution charcoal record is already published in Xiao et al. 2015 JQS as well as the pollen analyses. In addition to the high resolution charcoal record, the novelty concerns fire and vegetation interaction using fire episodes and frequency indexes compared to vegetation diversity indexes obtained from pollen assemblages from the same core. The authors reach the conclusion that fire occurs during cold and dry climate periods characterized by evergreen oaks and Alnus, and that fire lead to decrease in abundance of Lithocarpus/Castanopsis and tropical arbors. The material and methods section (and related figures) should be reduced as this is already published in details in Xiao et al. 2015 JQS. The discussion section needs to be restructured with a clear description of fire adapted and fire-sensitive taxa found in the region today in order to discuss fire impact on vegetation through time, and a discussion about how climate/monsoon drives the vegetation in the region should be presented before discussing the possible role of fire on the vegetation. Specific comments below should help to improve the manuscript in this way.

**Response:** We appreciate very much the referee's positive comments on the manuscript. According to these suggestions, we reduced the material and methods section and readjusted the discussion section. We address each comment with explanation as follows.

Specific comments:

1- Title: it seems from the reading of the title that the charcoal and pollen records are new – remove "based on charcoal and pollen records"

**Response:** Done. Thanks.

2- Line 12 page 1: compared with the pollen record – already published in Xiao et al 2015 JQS. Please modify to read "compared with major taxa and diversity indexes" or something similar.

**Response:** We change "compared with the pollen record" into "compared with major pollen taxa and pollen diversity indexes".

3- Line 10 page 2: the reference of Power et al. 2008 is wrong here

**Response:** Thanks. We change this reference into Zhao et al., 2005.

4- Line 21 page 2: "resulting in forest fire occurring frequently" please add some words here or in regional setting about the kind of vegetation that is burning today and the different fire adapted taxa and fire-sensitive taxa found in the region. This would help to follow the discussion.

**Response:** Relevant studies about the kind of vegetation that is burning today and the different fire adapted taxa and fire-sensitive taxa found in the region are very few. In here, we add some words "The study about forest fire during 1982-2008 in Yunnan Province shows that forest loss rates due to forest fire in different vegetation zones are different. The highest forest loss rate occurs in zone of semi-humid evergreen broadleaved forest dominated by *Cyclobalanopsis*

*glaucoides* and evergreen oaks and *Pinus yunnanensis* forest. Secondly, forest loss rate in zone of monsoon evergreen broadleaved forest dominated by *Castanopsis* and *Lithocarpus* is relatively low. Forest loss rate in zone of tropical rainforest and monsoon forest is lower than that in the former two vegetation zones."

At the same time, we add some words about vegetation regions around the study area in "Regional setting" section. "Qinghai Lake is located within zone of semi-humid evergreen broadleaved forest in Hengduan Mountains of western Yunnan Province, whose south is adjacent to zone of monsoon evergreen broadleaved forest of southwestern Yunnan Province (Wu et al., 1987)."

5- Line 25 page 2: would be good also to discuss macrocharcoal results and Black Carbon of the same core published in Palaeogeography, Palaeoclimatology, Palaeoecology, Volume 435, 1 October 2015, Pages 86-94 by Zhang et al.

**Response:** Ok, we add the discussion about macrocharcoal results and black carbon in the revised manuscript. Please see lines 24-30 page 7 and lines 1-3 page 8 in the revised manuscript for detail.

6- Material and Methods - section 3.1: the Table 1 and age model (Figure 3) are already published in details in Xiao et al. 2015 and in Zhang et al. 2015. Table 1 and Figures (3a and 3b) should be removed and references should be clearly indicated in the Material section 3.1. If the authors want to republish the age model, they should update their record from IntCal09 to IntCal13. For macroscopic charcoal analysis: please indicate that a low resolution record is already published in Xiao et al. 2015. Please explain in what the high resolution charcoal record will help to understand fire, vegetation and climate in this region.

**Response:** According to the suggestion, we remove Table1 and Figure 3 and cite the corresponding reference (Xiao et al., 2015). In the revised manuscript, we add an illustration about a low resolution charcoal record published in Xiao et al. 2015 and explain in what this high resolution charcoal record will help to understand fire, vegetation and climate in this region. Please see lines 26-30 page 2 in the revised manuscript.

7- Line 20 page 4: why standardizing pollen percentages? Please justify your approach.

**Response:** The reason is to see more clearly the interrelation among the variations of the selected pollen type percentages, we use min-max normalization to standardize pollen percentages and eliminate the influence of their values. We add an explanation in lines 2-4 page 5 in the revised manuscript.

8- Line 7 page 5: fire episode magnitude –discrepancy between units (particles/cm2) and the total charcoal influx?

**Response:** Thank the referee's careful observation. The unit of fire episode magnitude is particles/ cm$^2$/episode. We revise it in the revised manuscript.

9- Results – Chronology: remove this paragraph because this is already published in details in Xiao et al. 2015 – or shortened it and put this in the Material and Methods section – but clearly it is not new results. Keep the description of the temporal sampling of charcoal however.

**Response:** The paragraph about "Chronology" in the Results section is removed. The relative

results are shortened and put in the Material and Methods section. Please see lines 9-15 page 4 in the revised manuscript.

10- Section 4.2 Charcoal record, fire events and palynological diversity indices: this section is difficult to follow because the numbers reported in the text is not readable on Figure 4. Change the scale of the units reported in Figure 4. In addition, the authors describe the charcoal variations following pollen subzones determined in Xiao et al. 2015 but for pollen subzone TCQH-2 they changed this approach and determined subzone based on fire activity? As this paper is a focus on fire activity would be better to describe what is happening using fire activity zones and not following pollen subzones. Zone TCQH-4: only one fire event was detected – did the authors try several smoothing window and check whether this event remains in case of changing the smoothing background?

**Response:** The scale of the units reported in Figure 4 is changed. At the same time, Figure 4 is changed into Figure 2 according to other suggestions. Yes, pollen subzone TCQH-2 is determined based on fire activity in Xiao et al. 2015. According to this suggestion, we describe the results of charcoal record, fire events and palynological diversity indices using charcoal zones (fire activity zones). Please see lines 20-28 page 5, page 6 and lines 1-13 page7 in the revised manuscript. We try eight smoothing windows (300-1000 yr) and draw the conclusion that the fire event in Zone TCQH-4 (at 10.2 ka) remains in case of changing the smoothing background.

11- Line 22 page 7: "In the last 50 years, the charcoal concentration was still very low, and the relatively high CHAR may be at least partly due to the high sedimentation rate." Using the charcoal accumulation rate (CHAR) or in other word the charcoal influx instead of concentration is supposed to avoid "wrong" signal of fire in terms of concentration due to dilution. This sentence is unclear.

Discussion section: - Line 23 page 7: same comment as above about the sedimentation rate.

**Response:** Yes, CHAR reveals more real signal of fire than charcoal concentration. We rewrite this sentence, and make it clearer to detect signal of fire using CHAR. Please see lines 21-24 page 7 and lines 18-23 page 9 in the revised manuscript.

12- Line 16 page 8: While frequent fires appear to occur during the YD, it is unclear during the H1 giving the dates used by the authors (see sanchez Goni and Harrison 2010, QSR – HS1 is between 18 to 15.6 kyr cal BP) and the choice of using pollen zone to describe charcoal trend. In this case, low fire frequency is recorded during HS1.

**Response:** From our charcoal and fire activity records (Fig.2 in the revised manuscript), it can be seen that charcoal concentration, CHAR and fire frequency between 17.2 and 16.8 ka were relatively low, compared to their high values during the periods 18.5-17.2 ka and 16.8-15.0 ka. However, they were higher than most values in low value periods (15.0-13.0 ka, 11.5-4.3 ka, and after 0.8 ka). The major objective of this paper is to discuss stage changes of fire activity (such as during H1, BA, YD, HCO), and does not involve in more detailed changes in these stages. Thus, the period 17.2-16.8 ka with relatively low charcoal concentration, CHAR and fire frequency is included in the period 18.5-15.0 ka, and considered as one period with relatively high charcoal concentration, CHAR and fire frequency as a whole.

In the previously published study, the pollen record reveals that the climate during the period

17.9-15.0 ka corresponds to H1, and the climate during the period 18.5-17.9 ka was also cold. There are some deviations between the time of H1 in this study area and the result of sanchez Goni and Harrison 2010, QSR, which may be caused by regional difference or age uncertainties.

13- From line 29 page 8: Artemisia pollen percentages are also high in TCQH-1b. Why Poaceae and Artemisia would be indicative of human activities from 4.3 ka? "Superimposed influence of human activities and climatic cooling and drying" what are the proxies that indicate a cooling and drying then?

**Response:** *Artemisia* is a dry-tolerant herb, and sometimes a pioneer in cleared lands in the wooded mountains. Poaceae pollen, especially cereal type, is a common indicator of human disturbance or agricultural activity. Although *Artemisia* pollen percentages are also high in TCQH-1b, there is no other signal of human activity in this period. Single signal of high *Artemisia* pollen percenages can not indicate human activity. Whereas the increase in *Artemisia* pollen percentages from 4.3 ka accompanied with the rapid increase in Poaceae pollen percentages and the rapid degradation of the primary evergreen broadleaved forest, thus human activity may have influenced vegetation changes from 4.3 ka. Of course, the rapid degradation of the primary evergreen broadleaved forest revealed by the pollen record may be also influenced by climatic cooling and drying. The reasons are as follows. On the one hand, intensity of human activity at this period is not enough to make this abrupt change in vegetation; on the one hand, the other independent climatic proxies and the evidence of the cultural responses also show a clear climate drying between 4-5 ka (Zhao et al., 2009). Thus, we consider superimposed influence of human activities and climatic cooling and drying may result in abrupt changes of vegetation and fire activity at 4.3 ka in this study.

Zhao, Y., Yu, Z.C., Chen, F.H., Zhang, J.W., Yang, B. Vegetation response to Holocene climate change in monsoon-influenced region of China. Earth-Science Reviews 97 (2009) 242–256.

14- Line 9 page 9: again unclear about dilution and charcoal influx. Add the sedimentation rate curve to one of the figures.
**Response:** The sedimentation rate curve is added in Figure 2 (Figure 4 in the origin manuscript). At the same time, we rewrite this sentence, and demonstrate that dilution results in low concentration, not charcoal influx (CHAR). Signal of fire revealed by CHAR is more real than charcoal concentration. Please see lines 18-23 page 9 in the revised manuscript.

15- Line 13 page 9: "Paleofire studies at global scales reveal that high fire activity occurred during warm interstadials or interglacials, and low fire activity occurred during cold stadials or glacials (Power et al., 2008; Mooney et al., 2011; Marlon et al., 2013)". The reference of Mooney et al. 2011 is for the Australasia, regional scale. Add for interstadials and stadials Daniau et al. 2010 QSR. Marlon et al 2013 is for the Holocene only.
**Response:** According to the suggestion, we add a reference (Daniau et al., 2010 QSR), and delete the reference (Marlon et al 2013) because it is for the Holocene only.

16- Line 7-8 page 10: same comment about references
**Response:** Done.

17- section fire activity and vegetation: this section would benefit of a clear description of what are the fire adapted and fire-sensitive taxa found in the region today. In addition, it would be good to discuss how climate/monsoon drives the vegetation in the region before discussing the possible role of fire on the vegetation composition.

**Response:** This is a good idea to describe what are the fire-adapted and fire-sensitive taxa in the region. However, fire sensitivity studies of plant types in this region are very lacking, and there is still no special research in this region so far. At present, we can only analyze flammability of plant types according to forest loss rates due to forest fire in recent years in different vegetation zones. In "Introduction" section, we add these contents and consider that semi-humid evergreen broadleaved forest dominated by *Cyclobalanopsis* and evergreen oaks and *Pinus yunnanensis* forest are flammable, and monsoon evergreen broadleaved forest dominated by *Castanopsis* and *Lithocarpus* and tropical rainforest and monsoon forest are relatively nonflammable compared to semi-humid evergreen broadleaved forest and *Pinus yunnanensis* forest. Please see lines 21-26 page 2 in the revised manuscript.

We add relevant contents about how climate drives the vegetation in the region before discussing the possible role of fire on the vegetation composition. Please see lines 23-29 page 10 in the revised manuscript.

18- Figure 2 is already published in Xiao et al. 2015.
**Response:** This figure is deleted in the revised manuscript.

Technical details:
- Line 3 page 9 and others: "Edirotial Board" modify to read "Editorial"
**Response:** Done. Thank the referee's careful observation.

[revised manuscript text omitted]

---

## Author Response (AR1)

**Response to the Editor:**

**Editor Decision: Reconsider after major revisions** (13 Sep 2016) by Dr Nathalie Combourieu Nebout

Comments to the Author:

Dear authors,

I have carefully read your response to reviewers and had a look on what you proposed to modify your manuscript accordingly.

Reviewer 2 asked major revisions and I agree with this decision. You propose a collection of corrections but it appears that several of them are not sufficiently discussed or not enough detailed in the text. Please see the following comments and remarks.

**Response:**

Dear Editor Nebout,

   Thank you very much for your careful observation and comments. We revised the manuscript according to your comments and remarks. A point-by-point response to your comments and all relevant changes made in the revised manuscript are listed as follows. Please see the file "Response to the Editor" and the marked-up manuscript version for detail.

   We hope we successfully complied with the comments and remarks, and we appreciate your detailed reviews and suggestions.

Kind regards,

Xiayun Xiao

In name of all the co-authors.

1- The reviewer 2 ask you to remove figure 3 and not figure 2. I would like you conserve this picture. I think that the figure 2 is informative for local climate. However, it will be better to add this "ombrothermic" diagram directly on figure 1. It will have a sense as you show a map with the location of the site and the nearby towns. In this figure localities names and elevations values on the coloured map on the right are too small or too light and then unreadable. What is the little rectangle on the right and at the base of the map at left, it is completely unreadable.

**Response:** Thank you very much for your detailed comment. We conserve figure 2 and add this "ombrothermic" diagram directly on figure 1. The coloured map on the right of figure 1 has been modified. The little rectangle on the right and at the base of the map at left is the nine-dash line in the South China Sea, namely the traditional maritime boundary line of China. The black dots in the nine-dash line represent the islands in the South China Sea. In the revised figure 1, we add the words "islands in the South China Sea" in the little rectangle to illustrate meaning of the little rectangle. Please see the revised figure 1 for detail.

2- Reviewer 1 ask you about the concentration compared to percentages. This have to be discussed in the method. Your choices have to be described as they are the base of discussion. Perhaps you can show the figures proposed in your response as additional data and only justify your choices in the text with reference to the additional document

**Response:** According to the comment, we add a paragraph about selection of percentages or concentrations of the main pollen taxa in the method. The figures (Fig. SF1 and SF2) in this paragraph are served as an additional document, and it is cited in the text. Please see lines 6-18 page 5 in the marked-up manuscript version and Fig. SF1 and SF2.

3- As the two reviewers, I think that the relation between plant and fire sensitivity is not sufficiently developed (in the

revised version too). In fact you add some words but it remains insufficient as you use largely the opposition between fire-adapted and fire non-adapted plants or groups of plants. You have to explain more which taxa are fire sensitive and which are not in your record. Perhaps it will be good to add a paragraph on that when describing the vegetation or in the methods. This discussion has to be supported by references in my opinion. It is important to reinforce your paper discussion. You begin an explanation in the response but it is not developed in the corrected manuscript in my opinion. Your sentence about the vegetation of the Hengduan Mountain is not sufficient. What is requested is a detail about which taxa are expected to be burned preferentially and which not, based on observations (present day or past data). It will be good to use references such as Zhang et al 2015 before page 7 in this discussion.

**Response:** According to the comment, we add a paragraph to explain which taxa are fire sensitive and which are not in our record. Please see lines 3-20 page 4 in the marked-up manuscript version for detail.

We also add some discussion in the Discussion section according to these results of modern ecological studies on these main taxa in the record. Please see lines 20-31 page 13 and lines 19-25 page 14 in the marked-up manuscript version for detail. Accordingly, the conclusion is slightly changed. Please see lines 3-8 page 16 in the marked-up manuscript version for detail.

Some references such as Zhang et al 2015 have been cited in the Introduciton section. Please see lines 6-7 and 15-18 page 2 in the marked-up manuscript version for detail.

4- Concerning the standardization of your data, rephrase your sentence, it remains unclear. Explain more.

**Response:** We rephrase the sentence about the data standardization in the revised manuscript. Please see lines 4-7 page 6 in the marked-up manuscript version for detail.

5- You changed the zonation labelling. OK it was requested by reviewer 2. However the pollen zonation may be still traced on the figure I front to the charcoal one (used in the text) for a link to other papers even if it is not commented in the text (only mentioned in the figure with the reference to the published paper).

**Response:** Considering synthetically the comments of the editor and the reviewer 2, the pollen zonation (from TCQH-1 to TCQH-7) was added in the figure 2, which will make the link with the other published paper based on pollen data. At the same time, we add some words in the Result section to illustrate the relationship between charcoal zone and pollen zone. Please see the revised figure 2 and line 23 page 6, lines 1-3, 12, 20-23, 30 page 7, and lines 8, 16 page 8 in the marked-up manuscript version for detail.

6- As commented by the reviewer 2 there is difference in the time span of the H1 event in the studied record and the theoretical interval. The H1 has been defined in North Atlantic and you have to respect the chronology established there. If your event does not cover the whole interval or extend more or if you have differences and specificity in China, please explain more and discuss this fact.

**Response:** We added some discussion about the H1 in our study area. Please see lines 28-32 page 9 and lines 1-2 page 10 in the marked-up manuscript version for detail. Namely, we synthesized the results of Qinghai Lake, Tiancai Lake, and Lugu Lake located in different latitudes and altitudes in western Yunnan Province. Because of the different responses of the three study sites to global climate change and age uncertainty, the H1, B/A, and YD are considered to have occurred during the periods $17.5\pm0.5\sim15.4\pm0.4$ ka, $14.4\pm0.2\sim12.9\pm0.1$ ka, and $12.9\pm0.1\sim11.5$ ka in western Yunnan Province, respectively. The time of the H1 is almost consistent with the date defined by Sanchez Goñi and Harrison (HS1 is between 18 to 15.6 kyr cal BP) (Sanchez Goñi and Harrison, 2010).

The sentense "the frequent and intensive fire activity during the periods 18.5-15.0 ka and 13.0-11.5 ka corresponded roughly to the H1 and the YD." is not very accurate. We modified it into "the frequent and intensive fire activity during the periods 18.5-15.0 ka and 13.0-11.5 ka corresponded to the relatively cold period before the H1 and the H1, and the YD, respectively." Please see lines 12-13 page 10 in the marked-up manuscript version for detail.

Sanchez Goñi, M. F. and Harrison, S. P.: Millennial-scale climate variability and vegetation changes during the Last Glacial: Concepts and terminology, Quaternary Sci. Rev., 29, 2823–2827, 2010.

7- I agree with reviewer 2 about Artemisia and Poaceae significance. If you interpret as human influence, you have to justify more. Have you Cereals pollen since 4.3kyr? if not Poaceae occurrence even in high percentage do not be explained by human influence and Artemisia is then considered as expressing climate response of the vegetation. Perhaps you have other arguments to assess human impact sinde 4.3 kyr.

**Response:** It is difficult to separate cereal pollen from Poaceae pollen. Based on studies of the characteristics of modern Poaceae pollen, Poaceae grains with diameters greater than 40 μm were usually classified as cereal-type (Lamb et al., 2003; Huang and O'Connell, 2000). In our study, Poaceae grains with diameters greater than 40 μm began to occur at 18.5 ka and its pollen percentages had no obvious change (Fig. SF3), which didn't reflect changes of human activity. Thus, the criterion of cereal pollen (Poaceae grains with diameters greater than 40 μm) may be not applicable in this study area. Studies on the characteristics of modern Poaceae pollen in Yunnan Province are needed for redefining the criterion of cereal pollen for this study area.

Because cereal pollen was not separated from Poaceae pollen, we can not still explicitly declare that increase of Poaceae pollen percentages is an indication of human activity. Thus, we deleted the sentence "Pollen indicative of human activities (e.g., Poaceae and *Artemisia*) unambiguously increased from 4.3 ka.". However, because cereal pollen was not separated from Poaceae pollen, we can not rule out the possibility of increase in human activity when Poaceae pollen percentages increased. At the same time, combined with the other signals in the pollen assemblage (such as the rapid degradation of the primary evergreen broadleaved forest) and the archaeological records (such as Neolithic culture began to emerge at around 4 ka or a little earlier), we speculate that superimposed influence of human activities such as forest clearance and agricultural cultivation and climatic cooling and drying may result in the frequent and strong fire activity between 4.3 and 0.8 ka.

Thus, we add these contents in the revised manuscript. Please see lines 25-32 page 10 and lines 1-6 page 11 in the marked-up manuscript version for detail.

Lamb, H., Darbyshire, I., Verschuren, D.: Vegetation response to rainfall variation and human impact in central Kenya during the past 1100 years, The Holocene, 13(2), 285-292, 2003.

Huang, C.C., and O'Connell, M.: Recent land-use and soil-erosion history within a small catchment in Connemara, western Ireland: evidence from lake sediments and documentary sources, Catena, 41, 293–335, 2000.

[Figure]

Fig. SF3 Pollen percentages of Poaceae (>40 μm) since 18.5 ka in Qinghai Lake.

8- The reference of Marlon has not been deleted in the bibliography of the manuscript I received.

**Response:** Thank you very much for your careful observation. We deleted the reference of Marlon in the bibliography of the revised manuscript.

This paper is certainly interesting and brings new approach of the data presented within the text and figures. However I

would welcome a new version implemented accordingly to the additional comments and corrections that I request and a reply to my remarks listed above. I would probably send the paper back to other reviewers on receipt.

sincerely yours

Nathalie Combourieu-Nebout

**Response:** We appreciate very much your positive comments on the manuscript. According to your comments and remarks, we revised carefully the manuscript and address each comment with explanation as above. Please see the file "Response to the Editor" and the marked-up manuscript version for detail.

We hope we successfully complied with the comments and remarks, and we appreciate your detailed reviews and suggestions.

[revised manuscript text omitted]

---

## Author Response (AR2)

**Response to the editor and the reviewers:**

**Major Revision**
**Editor Decision: Reconsider after major revisions** (06 Dec 2016) by Dr Nathalie Combourieu Nebout
Comments to the Author:
Dear authors,

We have now received new reports from two reviews concerning your paper. Both reviewers (especially two of them) made important comments about your manuscript that need to be absolutely taken into consideration in a new version. Particularly one of the reviewers would like to have more details about your hypothesis about diversity and to develop what you want to demonstrate with this curve, the dynamic of diversity in the studied area and the comparison with the other data. Precision and adjustments are also needed about fire effect on the different component of the vegetation and it is necessary to amend the discussion on fire too.

I am strongly recommending you to reply quickly to both reviews, justify your response to all comments and follow the reviewers suggestions and remarks.

Please prepare a revised version of your paper accordingly. In the revised version, I would like to see your corrections in track change mode. Please, don't forget to send, with your revised version, a replies to all the comments explaining how you have modified your manuscript. With the new manuscript and your response, I will decide if your paper will need again other reviews.

Waiting after your new version
Best regards
Nathalie Combourieu-Nebout

**Response:**
Dear Editor Nebout,
      Thank you very much for your comments and your help. We revised carefully the manuscript according to the reviewers' comments. Especially, we proposed a hypothesis that effects of fire on biodiversity in different regions may be different in the introduction. The discussion section was reorganized, and more discussion about the dynamic of diversity was added in the discussion. Fire effect on the different component of the vegetation was more precisely illustrated. A point-by-point response to the reviewers' comments and all relevant changes made in the revised manuscript are listed as follows. Please see the file "Response to the editor and the reviewers" and the marked-up manuscript version for detail.
      We hope that we successfully complied with the comments, and this revised manuscript can meet the requirements. We appreciate your help again.

Kind regards,
Xiayun Xiao
In name of all the co-authors.

**Report #1**

Comments on "Postglacial fire history and interactions with vegetation and climate in southwestern Yunnan Province of China" by Xiayun Xiao et al.

This paper utilizes a traditional method, charcoal analysis, to study the linkage among wildfires, vegetation, and climate change. Although this topic is not new, considering the sparseness of fire history in Southwestern China, a region prone to drought, such study is still welcome. This paper is well-organized, and it presents information on the flammable, fire-adapted, and fire-tolerant plants. It should be published with more audiences. But some statements seems imprecise. I point out some of, but not limited to, them listed below. I recommend the paper to being published with minor revisions.

**Response:** We appreciate very much your positive comments on the manuscript. According to your comments, we revised carefully the imprecise statements in the manuscript and address each comment with explanation as follows. Please see the following response and the marked-up manuscript version for detail.

1. Abstract. "Vegetation responses to fire before 4.3 ka are not consistent with that after 4.3 ka" I suggest changing it to" Vegetation responses to fire aftere 4.3 ka are not consistent with that before 4.3 ka".

**Response:** Done. Please see lines 18 page 1 in the marked-up manuscript version for detail.

2. Lines 11-12, Page 2, "At present, studies on fire history and the interactions between long-term vegetation, fire, and climate are mainly concentrated in the world's boreal forests (Zhao et al., 2005)." I think that this statement maybe not precise. Now there is a Global Charcoal Database (https://www.ncdc.noaa.gov/paleo/impd/gcd.html), which summarized the wildfire data worldwide. You can check the database.

**Response:** We check the database, and find that most sites on paleofire concentrate in North and South America. Thus, we change "At present, studies on fire history and the interactions between long-term vegetation, fire, and climate are mainly concentrated in the world's boreal forests (Zhao et al., 2005)." into "At present, studies on fire history and the interactions between long-term vegetation, fire, and climate are mainly concentrated in North and South America (https://gis.ncdc.noaa.gov/map/viewer/#app=cdo&cfg=paleo&theme=paleo&layers=000000000001)."

3. Lines 6-7, Page 2. "Recently, black carbon was also gradually used as evidence of fire history reconstruction (Schmidt and Noack, 2000; Wang et al., 2013; Wolf et al., 2014)." I would like to suggest adding "black carbon was also gradually used as evidence of fire history reconstruction (Schmidt and Noack, 2000; Wang et al., 2013; Wolf et al., 2014) and its subtypes of char and soot (Han et al., 2012, Global Biogeochemical Cycles; Han et al., 2016, Sci. Rep.) have potential to differentiate between smoldering and flaming fires."

**Response:** Done. Please see lines 17-18 page 2 in the marked-up manuscript version for detail.

4. Page 2, "In southwestern China, there is only one study that reconstructed fire history by using black carbon (Zhang et al., 2015b)." I would like to suggest adding "to the best of our knowledge".

**Response:** Done. Please see lines 26 page 2 in the marked-up manuscript version for detail.

5. Page 2 " These studies on fire history in China have concentrated on the relationship between fire activity and climatic change on orbital to suborbital timescales, whereas studies of past fire frequency and magnitude and its relationship to climate, human activity and vegetation dynamics have rarely been attempted in China." References are needed. Also, I don't think that this statement is precise. I agree with you that there are limited data on historical wildfire reconstruction. But I think that there are still some studies on " past fire frequency and magnitude and its relationship to climate, human activity and vegetation dynamics".

**Response:** We further searched studies on fire history in China, and found an important reference which analyzes the interaction between climate, fire, and vegetation history over the last 2000 years in southwestern China. Thus, we changed "whereas studies of past fire frequency and magnitude and its relationship to climate, human activity and

vegetation dynamics have rarely been attempted in China." into "Studies of past fire frequency and magnitude and its relationship to climate, human activity and vegetation dynamics are however, uncommon for China." At the same time, some references are added. Please see lines 30-31 page 2 in the marked-up manuscript version for detail.

6. Line 23, Page 2. " The study about forest fire during 1982-2008 in Yunnan
Province" Please give a ref.
**Response:** A reference was cited. Please see lines 4 page 3 in the marked-up manuscript version for detail.

7. Lines 20-23, Pages 10. Regarding to the 8.2 event with no corresponding wildfire event, I would like to suggest considering the temperature influence. General speaking, low temperature would inhibit wildfire occurrence, and this can also be reflected from the low wildfire intensity in Daihai, China (Han et al., 2012, GBC), where there has a far lower temperature due to its location in north China. In Asian monsoon-influenced area, the dryness and temperature are two competition factors influencing wildfire activities. In my opinion, dryness is very likely to be the key factor influencing fire occurrences, but the temperature factor cannot be ignored. Or else, it seems that cold climate facilitates fire occurrences. But this apparently is not the case from modern observations. This point should be clarified in this paper.
**Response:** Yes, dryness is the key factor influencing fire occurrences in southwestern China, but the temperature factor cannot be ignored. Thus, we add influence of the temperature on fire occurrences when explain the reasons for the abrupt cold event at 8.2 ka not marked in the charcoal record. Please see lines 6-9 page 11 in the marked-up manuscript version for detail.

**Report #2**

The study by Xiao et al. deals with fire dynamics in comparison with vegetation proxies since the LGM. Fire regime is reconstructed from a high resolution macrocharcoal record while vegetation is reconstructed from the same core using pollen percentages and diversity indices calculated from the pollen record. The data are then compared and discussed based on visual observation of those tendencies. Those comparisons are then discussed and compared to pollen from other lakes and a d18O record. This record is cited only once in the text in the discussion section, and I wonder if it is really useful in the manuscript. The text is well written but will require copyediting from a native English speaker, for example multiple occurrences of the word 'arbor' thorough the text is confusing, do the authors mean trees? That being said the manuscript in its present form lacks some concepts that will be useful to strengthen its message and to clearly demonstrate the novelty compared to Xiao et al. 2015 that is roughly based on the same data.
**Response:** Though the $\delta^{18}O$ record was cited only once in the manuscript, it represents a regional climate in the Indian Ocean. So we consider it is useful in this manuscript. In our manuscript, the word 'arbor' means trees and shrubs. We added an explanation of 'arbor' following the word in the revised manuscript, please see lines 14 page 4 in the marked-up manuscript version for detail. The revised manuscript was copyedited by Ph. D Mark Burrows (a native English speaker), because Professor Simon G. Haberle (the second author) is still on holiday.

The use of diversity indices is interesting and I more than agree that diversity from a paleo-perspective is capital for understanding past ecosystem dynamics. However, those indices are underexploited in the manuscript, removing them from the analyses and figures will not significantly change the manuscript and its message. What are the hypotheses about diversity that the authors are testing? Those hypotheses are lacking in the introduction; for example, do the authors expect diversity to increase or decrease after fires, do they expect intermediate disturbance promoting higher diversity? What relationship between evenness and richness is expected? The discussion about diversity dynamics could be interesting but is limited in the discussion and few conclusions are drawn for diversity.
**Response:** Thank the reviewer's positive comment on the significance of our manuscript. At present, there is still argument about effects of fire upon biodiversity. Namely, some ecologists have discovered that fire suppression (or the

exclusion of fire-catalyzed practices) is a powerful destroyer of biodiversity (Pyne, 1998). On the other hand, some studies suggest that plant diversity increased after fires (Trabaud and Lepart, 1980; Stocker, 1981; Nasi et al., 2002). Thus, we put forward the opinion that effects of fire on biodiversity in different regions may be different. In order to make a reasonable strategy for forest fire management for a region, we need to better understand effects of fire upon biodiversity and driving mechanism of forest fires in this region. These may be achieved through long-term fire history research (Tinner et al., 1999; McWethy et al., 2013; Morales-Molino et al., 2013; Kloster et al., 2015). We add these contents in the introduction. Please see lines 29-31 page 1 and lines 1-5 page 2 in the marked-up manuscript version for detail.

In the discussion, we added some relatively discussion about diversity and drew a conclusion about effects of fire upon palnt diversity. Namely, plant diversity increased after fires. Based on this conclusion, we consider that fire suppression and control is reasonable in our study area. please see section 5.6 in the marked-up manuscript version for detail.

We illustrated notions of evenness and richness in the section 3.3 of the origin manuscript. Following this illustration, the relationship between evenness and richness is explained in the revised manuscript. Please see lines 3-6 page 6 in the marked-up manuscript version for detail.

The discussion about diversity dynamics are added, please see section 5.6 (lines 9-31 page 15 and lines 1-13 page 16) in the marked-up manuscript version for detail.

The manuscript in its present form is thus too descriptive and lack hypotheses. This is also apparent in the treatment and the discussion about fire effect on the vegetation. The discussion about Alnus is particularly confusing. The authors seem to get lost in conjectures for explaining the abundance of Alnus concomitantly with fires. Clearer hypotheses on fire effect on vegetation will make the discussion more objective. Fire can act by promoting fire adapted species (serotinous trees for examples) or fire resistant species (resprouting trees and shrubs) and early successioners. Alnus are not flammable and the results of the analysis do not support Alnus flammability: trying to demonstrate that Alnus patch can burn under various fire regime cannot prove that they are flammable.

**Response:** We deleted some descriptive contents (such as P13 l7-16 and P15 l15-9) in the discussion. In the revised manuscript, we confirmed that *Alnus* are nonflammable taxa and revised the discussion about the abundance of *Alnus* concomitantly with fires. Please see lines 16-20 page 14 in the marked-up manuscript version for detail.

The authors indicate at numerous occasions that they are measuring fire intensity, this is not possible using sedimentary charcoal records. Fire intensity is in W.m2 and the relationship between charcoal number and fire intensity has to be demonstrated. Likewise fire episode magnitude should be avoided because a single charcoal peak can encompass multiple fire episodes, also it is unclear what this measure is referring to: intensity, severity, burned area, burned biomass. Most of the studies are agreeing that peak magnitude and raw charcoal values are indicative of burned biomass.

**Response:** Thank you very much. This is a very professional comment. We looked up some references (Keeley, 2009; Ali et al., 2012) and understood that fire intensity describes the physical combustion process of energy release from organic matter, thus it is justifiably restricted to measures of energy output and it has the units of $W\ m^{-2}$. However, fire intensity is sometimes used incorrectly to describe fire effects. Fire severity has generally emphasized degrees of organic matter loss or decomposition both aboveground and belowground, which is positively correlated with fire intensity. Thus, we changed "intensity" to "severity" in the revised manuscript.

Considering that a single charcoal peak encompasses multiple fire episodes, we change "fire episode magnitude" to "peak magnitude". The magnitude of CHAR peaks can be used as a proxy for fire severity (Ali et al., 2012), indicating further degrees of organic matter loss or burned biomass.

Keeley JE (2009) Fire intensity, fire severity and burn severity: A brief review and suggested usage. Int J Wildland Fire 18(1):116–126.
Ali AA, Blarquez O, Girardin MP et al. Control of the multimillennial wildfire size in boreal North America by spring climatic conditions. PNAS, 2012, 109(51): 20966-20970.

I carefully read the author response to the two first reviewers, if most of the previous comments are very well addressed, I still question what is the marker of human influence. For example, p10 line 4 the authors argue that the climate is accompanied by signs of slash and burn and cite Xiao 2015. What are those signs? In Xiao 2015 the slash and burn is emitted as a hypothesis in the discussion, and here this hypothesis appears as a certitude. How can the authors prove that?

**Response:** We changed this certain opinion in the origin revised manuscript into a hypothesis. Namely, we changed the sentence "After 3.8±0.5 ka the climate became cooler and drier, accompanied by signals of human activity such as slash and burn (Xiao et al., 2015)." to "After 3.8±0.5 ka the climate became cooler and drier, which may be accompanied by signals of human activity such as slash and burn (Xiao et al., 2015)." Except for this sentence, human influence was deduced as a hypothesis in the other places in the origin revised manuscript. The reasons for possible human influence from 4.3 ka were listed in lines 1-4 page 11 in the origin revised manuscript (lines 9-23 page 12 in the marked-up manuscript version).

The discussion could by shortened significantly without losing information, large parts of the discussion are actually results and should be moved in the appropriate section. The discussion could also be reorganized to become clearer and less descriptive, for example the 5.1 paragraph deal with too numerous topics such as fire climate and human and is difficult to follow.

**Response:** The discussion was reorganized. Some descriptive contents (such as P13 l7-16 and P15 l15-9) are deleted. Please see the discussion in the revised manuscript for detail.

I hope that those comments will help the authors to strengthen the message and hypotheses of their study.

**Response:** Thank you very much for your constructive comments. After the manuscript was revised according to these comments, we considered that the quality of the revised manuscript has been evidently improved.

Specific comments:

P1 l13 'correlated' imply that the authors did some sort of correlation analysis, replace with related for example

**Response:** Done.

P1 l11 intensity is not well measured from charcoal records, authors can have a look at Ali et al. 2012 PNAS for size/biomass burning/frequency reconstruction

**Response:** We read carefully Ali et al's (2012) and Keeley (2009)'s studies, and understood difference of the terminology of fire intensity and fire severity. Namely, fire intensity is justifiably restricted to measures of energy output, whereas fire severity has generally emphasized degrees of organic matter loss or decomposition both aboveground and belowground, which is positively correlated with fire intensity. Thus, we change "intensity" to "severity" in the revised manuscript.

P1 l20 Alnus are not flammable plants

**Response:** We revised it in the revised manuscript.

P1 l22 and elsewhere specify 'abor' meaning

**Response:** The word 'arbor' means 'trees and shrubs'. We added an explanation of 'arbor' following the word when it first appeared.

P2 l8 Ali et al. 2009 is not clearly about management, but the authors may have a look at Oris et al 2014 Ecol Applications.

**Response:** We read carefully the paper (Oris et al., 2014), and put forward some corresponding suggestions for management strategy for forest fires to our study area. Please see lines 14-30 page 16 in the marked-up manuscript version for detail.

P2 l12 Zhao et al citation is not about boreal forest. Numerous examples from the boreal forest exist

**Response:** According to the first reviewer's comment, we revised the sentence and its citation. Please see lines 20-21 page 2 in the marked-up manuscript version for detail.

P3 l16 '1990', did the catchment area changed since 1990, why does using a date here be useful?

**Response:** No, the catchment area didn't change since 1990, but the surface area of the lake changed because of the change of water level. We changed this sentence into 'Qinghai Lake is a volcanic dammed lake with a catchment area of 1.5 km$^2$ and a surface area of c. 0.25 km$^2$ (in 1990), respectively (Wang et al., 2002).'

P5 and 6 it is not clear how palynological richness was calculated, in the result section richness is given with decimals, if the raw number of taxa is used where those decimals are coming from? If raw number of taxa is used from pollen counts, those results are not reliable because of species area type relationship. The richness need to be rarefied to the same pollen count, authors should have a look at Birks and Line 1992 The Holocene. It is also unclear if data is transformed before or after evenness calculus.

**Response:** Palynological richness index is the number of different pollen types in every pollen sample, which was illustrated in the section 3.3 of the origin manuscript. The reason for richness given with decimals in the result section is that it is an average of palynological richness for every zone. Considering palynological richness index is an integer, we rounded average of palynological richness index in the revised manuscript.

We read carefully the reference (Birks and Line 1992 The Holocene). In the revised manuscript, rarefaction analysis was used to estimate palynological richness, which was computed using the vegan package in the R programming environment (http://CRAN.R-project.org/package=vegan) (Oksanen et al, 2016). Fortunately, their results and tendencies of rarefied richness and pollen taxa sum are very similar (Fig. S1).

[Figure]

Fig. S1 Comparison of the rarefied richness and the pollen taxa sum.

Data isn't transformed before and after evenness calculus. Simpson's reciprocal index (1/D) is calculated according to the formula ( $\dfrac{1}{D} = \dfrac{N(N-1)}{\sum\limits_{i=1}^{S} n_i(n_i-1)}$ ).

Oksanen J., Blanchet F.G., Friendly M., Kindt R., Legendre P., McGlinn D., Minchin P.R., O'Hara, R.B. Simpson G.L., Solymos P., Stevens M.H.H., Szoecs E., Wagner H. (2016) vegan: Community Ecology Package. R package version 2.4-1.

P6 It is unclear how the zonation was done, if based on fire frequency or fire events, why zone 2 has only 1 fire at the beginning and two at the end, it would have been more logical to define a zone without fires. If the zonation is based

on a calculus (distance metric?) or on visual estimation or any method the procedure used must appear in the method section.

**Response:** In order to facilitate a comparison between fire activity and major pollen taxa and previous vegetation reconstruction for the same core (Xiao et al. 2015), the zonation was done based on the visual inspection of the macroscopic charcoal and fire activity records and referring to palynological zonation boundaries. We added an illustration about the zonation in lines 9-10 page 7.

P8 l20 and elsewere "increased significantly" what test did the author used for assessing the significance of the change?

**Response:** Here, meaning of "significantly" is "evidently" or "markedly". It is replaced by "evidently" or "markedly" in the revised manuscript.

Result and discussion section: how do the authors interpret the concomitant increase in richness and decrease in diversity?

**Response:** The explanation about the concomitant increase in richness and decrease in diversity is listed in lines 4-11 page 16 in the marked-up manuscript version.

P10 l16-17 is seems to me that this is an over interpretation of the results.
**Response:** This sentence was deleted.

P10 l20 'was'
**Response:** Done.

P10 l4 circular reasoning see above
**Response:** Done.

P12 l3-6 What are those fires #1 and #2? How do those results relate to the present study?
**Response:** The fires #1 and #2? are two fire events inferred from the two charcoal peaks, occurred at 5120±66 cal. yr B.P. (fire event 1) and 1288±8 cal. yr B.P. (fire event 2) (Jiang et al., 2008). Jiang et al. (2008) consider that fire event 1 was a natural origin and probably facilitated by drying environmental conditions, whereas fire event 2 was caused by clearing and may be related to the spread of the Han farming culture. The relationship between these results and our study is that fires are all controlled by dry climatic conditions in the two regions under the background of natural factor' control.

P12 l19-24 Citations are missing
**Response:** A reference is cited.

P13 l7-16 and P15 l5-9 Move to results
**Response:** These sentences are deleted.

[revised manuscript text omitted]

---

## Author Response (AR3)

**Major Revision**

**Editor Decision: Reconsider after major revisions** (13 Mar 2017) by Dr Nathalie Combourieu Nebout
Comments to the Author:

Dear Authors,

We have received the report of one reviewer that asks you other corrections after having a look on your corrected version.
Please take carefully into account the remarks and corrections and send to Climate of the Past your new corrected manuscript with amendments in track change mode.

I will take my decision after looking on the revised version and decide if you paper will need supplementary reviews.
Best regards

Nathalie

**Response:**
Dear Editor Nebout,
    Thank you very much for your help. We took carefully into account the referee's remarks and corrections, and revised carefully the manuscript according to these comments. A point-by-point response to the referee's comments and all relevant changes made in the revised manuscript are listed as follows. At the same time, the second co-author (Professor Simon G. Haberle, a native speaker) carefully copyedited the final revised manuscript and highlighted all relevant changes. Please see the files "Response to the comments" and "the new corrected manuscript with amendments in track change mode" for detail.
     We hope that we successfully complied with the comments, and we would be grateful if you would consider this revised ms for publication in Climate of the Past.

Kind regards,
Xiayun Xiao
In name of all the co-authors.

**Report #1**

accepted subject to minor revisions

Comments on "Postglacial fire history and interactions with vegetation and climate in southwestern Yunnan Province of China" by Xiayun Xiao et al.

I carefully read the revised version of the manuscript by Xiayun Xiao et al. and their response to my previous comments. The authors did take my comments into account very well and I think that the manuscript in its present form is improved. I notably appreciated the calculation of rarefied richness, which obviously did not change significantly the diversity trend but is, nonetheless, mathematically correct in a paleoecological context. Following my reading I think that the manuscript could go into the interactive review process after some minor corrections listed below:

**Response:** We appreciate very much these positive comments on our revised manuscript.

1) I still did not agree with the use of the term "arbor" since its use in an ecological context appears awkward to me.

From the Cambridge dictionary I obtain "a sheltered place in a garden formed by trees and bushes that are grown to partly surround it: a rose arbor" and "An axle or spindle on which something revolves." From the Oxford dictionary. I would suggest to stick with "trees and shrubs".

**Response:** Ok, we replace "arbor" with "trees and shrubs" in the revised manuscript.

2) "Namely, plant diversity increased after fires. Based on this conclusion, we consider that fire suppression and control is reasonable in our study area. " I would take extreme caution regarding this and management scenario such as l30 p23. I do not know the study region the authors are dealing with, but extreme management of fire such as active suppression could have major consequences (e.g. 1986 yellowstone fire in the US that was the result of almost a century of fire suppression), not only on biodiversity but on human livelihoods, economy, etc.

**Response:** Ok, we changed this certitude into a hypothesis in the revised manuscript. Namely, we changed "Thus, fire control and suppression is a necessary management strategy for forest fires in the study area, which will reduce the risk of large forest fires and even keep relatively high biodiversity in the future." into "In our study area, fire is not favorable to plant diversity, and if it is of high enough frequency and intensity, fire may also deplete plant diversity. Thus, fire control and suppression may be one management strategy to maintain plant diversity in the study area, which will have the added benefit of reducing the risk of large forest fires and maintaining relatively high biodiversity in the future." Please see lines 24-27 page 20 in the marked-up manuscript version for detail.

3) "*Alnus* is a nonflammable tree species. However, if *Alnus* is used to prevent fires, it is necessary to plant pure Alnus forest, because fire resistant ability of low-density Alnus forest or mixed-Alnus forest is relatively weak." I think this sentence is not necessary and not related to the present study.

**Response:** We have rewritten the section to highlight the importance of *Alnus* as a fire resistant plant to be used in the management if fires in the region. The section is important to the paper as it addresses directly the management options that might be available to those who work to protect the region from major fires. The palaeoecological record can help to inform managers of the risk of major fires and so we would like to keep this section in the paper. Please see lines 15-18 page 20 in the marked-up manuscript version for detail.

4) "At present, studies on fire history and the interactions between long-term vegetation, fire, and climate are mainly concentrated in North and South America (https://gis.ncdc.noaa.gov/map/viewer/#app=cdo&cfg=paleo&theme=paleo &layers =000000000001). Based on these studies, the USA, Canada and parts of Europe have been actively advancing methods and numerical approaches to reconstruct long term fire histories. " those two sentence do not make sense: Fire reconstructions are not concentrated in NA and SA, please refer to the Global Charcoal Database if needed https://www.paleofire.org/index.php?p=CDA/index&gcd_menu=CDA (also the NOAA link is not working). Fire reconstruction is active outside of US Canada and parts of Europe: e.g. Australia, etc.

**Response:** I don't know why the NOAA link can be not opened now. We reopen the link (https://www.ncdc.noaa.gov/data-access/paleoclimatology-data/datasets/fire-history), and then click "Interactive Map", we can get a map of study sites on paleofire (Fig. S1). From this map, we find that study sites in North and South America seems to be denser than that in other regions.

[Figure]

Fig. S1 A map of study sites on paleofire from the NOAA link

According to the referee's comment, we open the Global Charcoal Database (https://www.paleofire.org/index.php?p=CDA/index&gcd_menu=CDA), then get another map of study sites on paleofire (Fig. S2). From this map, it seems to be that study sites on paleofire are mainly concentrated in North and South America, parts of Europe and Oceania.

[Figure]

Fig. S2 A map of study sites of paleofire from the Global Charcoal Database

Because the Global Charcoal Database includes more study sites on paleofire than the NOAA link, we adopted the result of the Global Charcoal Database. Thus, we changed these two sentences into "At present, studies on fire history and the interactions between long-term vegetation, fire, and climate are mainly concentrated in North and South America, parts of Europe and Oceania (https://www.paleofire.org/index.php?p=CDA/index&gcd_menu=CDA).

Based on these studies, methods and numerical approaches to reconstruct long term fire histories have been actively advanced." Please see lines 20-22 page 6 in the marked-up manuscript version for detail.

5) l21 p22 "destroyer" please use appropriate term
**Response:** We changed "be a destroyer of" into "degrade". Please see line 19 of page 19 in the marked-up manuscript version for detail.

6) l16 p22 "concentricity" please explain
**Response:** This sentence has been removed as we consider that it is now redundant and the point of how the richness and evenness of the forest responds to fire has already been adequately explained in this paragraph.

7) P22 and elsewhere "warmer species" do you mean species adapted to warmer climate?
**Response:** Yes. We changed "warmer species" into "species adapted to warmer temperatures". Please see line 18 of page 19 in the marked-up manuscript version for detail.

[revised manuscript text omitted]

---

## Author Response (AR4)

**Response to the Editor:**

**Editor Decision: Reconsider after major revisions** (07 Apr 2017) by Nathalie Combourieu Nebout
Comments to the Author:

5  Dear authors,

Thanks you for your documents. I have read carefully the revised version of your paper and your response to the reviewer
It seems that you follow the comments and corrected accordingly.

10  However I have now some questions that your new document generate when I read it again. Moreover I request some modifications in the text and figures :
**Response:**
Dear Editor Nebout,
    We appreciate very much your positive comments on our revised manuscript. We also thank you very much for

15  your careful observation and detailed comments, which further improves the quality of our paper. We revised the manuscript according to your comments and remarks. A point-by-point response to your comments and all relevant changes made in the revised manuscript are listed as follows. Please see the file "Response to the Editor" and "the marked-up manuscript version" for detail.
    We hope that we successfully complied with the comments, and we would be grateful if you would consider this

20  revised ms for publication in Climate of the Past.

Kind regards,
Xiayun Xiao
In name of all the co-authors.

1- First, the figures needs to be amended because the characters' size used is definitively too little? Take care to that because your figure has to be readable in 100% format and probably in a portrait presentation.
On Figure 1, could it be possible to place the onbrothermic diagram out of the map and enlarge its size, it is not possible to read it now in the figure. And do not forget to use bigger characters for the names and legend. In your

30  revised manuscript the figure as in black and white and in the first version it was colored. Finally what do you choose? Probably it will be better in colored format to better see the different elevation with a color panel from white to black
**Response:** According to this comment, we placed the onbrothermic diagram out of the map and enlarged its size, and used bigger characters for the names and legend. Finally, we chose this figure as in colored format to better see the different elevation. Please see the revised Figure 1 for detail.

35

On Figure 3 Could you please indicate at the top what corresponds to your data, what originate from other site and what is from Arabian Sea?
On the same figure, I do not understand what you mean with the colored band as they are not in phase. If it corresponds to the climate phases of the deglaciation and Holocene, defined by the isotope curves (H1, B/A, …) it

40  appears strange to have different width for the different curves as the duration of these periods have been defined in several papers from isotopes and ice cores results (for example Björk et al;, 1998 and Walker et al, 2012 and others like those you have cited).
**Response:** We added the origin of the data at the top of Figure 3, and climatic phases defined in several papers cited by this paper on the right side of Figure 3.

45

In addition, you compare with the Arabian Sea. OK I am not a specialist of Asian region but there are probably other marine records nearest from your site that show this period in detail. Justify why you choose that one instead of nearest

ones.

**Response:** Yes, there are some other marine records nearest from our site, for example the $\delta\,^{18}$O record from core KL126 in the Bay of Bengal (Kudrass et al., 2001) and the sediment colour record from northeastern Arabian Sea (Deplazes et al., 2013) cited in this paper. The reason why we choose the $\delta\,^{18}$O records from core NIOP905 in the western Arabian Sea instead of nearest ones is that it can show that this climatic trend is similar in a larger area.

In the supplementary data please make the names in SF 1 and 2 and ages especially ages in SF2. Perhaps you may delete one age label per two and enlarge the characters.
**Response:** Done.

2- Page 14. Could you please define the acronyms DCA and PCA. Lot of people know but when you introduce an acronym it is better to define it the first time. In addition you talk about these mathematical analyses without any presentations. I know that it corresponds to previous study but does it bring anything in the discussion? You compare DCA in your core to PCA in another core. May be more explanation are needed to reinforce your text. One or two sentences only will be necessary.
**Response:** The acronyms DCA and PCA were defined in line 3 page 14 and line 6 page 14 of the marked-up manuscript version, respectively. We added two sentences "Selection of the DCA or PCA in different studies is based on the underlying linearity of the data. Namely, the PCA is selected when most of the underling responses are linear or at least monotonic to the underlying latent variables, otherwise the DCA is selected (Xiao et al., 2014a)." to explain how to select the DCA or PCA in lines 8-10 page 14 of the marked-up manuscript version.

3- Page 17-18. Your discussion about flammability of Quercus is very interesting. Did you try to determine the macrocharcoals? I do not know if it is possible but I think it will help you too. There is no real conclusion at the end of this paragraph. What did you conclude for your site? Which hypothesis did you choose to interpret the Quercus curve at your site? Perhaps it is the organization of the paragraph some conclusion are before the hypothesis proposed by other authors.
**Response:** Macroscopic charcoal indicates fire activity. According to flammability of evergreen oaks, we can speculate that semi-humid evergreen broadleaved forest dominated by evergreen oaks in this study area is prone to fire and may produce more charcoals, which is helpful to management strategy for forest fires. We have discussed implication of evergreen oaks for management strategy for forest fires in the 5.7 section (lines 18-22 page 20 of the marked-up manuscript version).
   According to the relationship between fire activity and pollen percentages of evergreen oaks, we conclude that evergreen oaks are flammable plants in line 27 page 17 and they are relatively fire-tolerant taxa in line 2 page 18 of the origin manuscript. Maybe it is not very clear. We add this conclusion before the hypothesis proposed by other authors. Please see lines 5-6 page 18 in the marked-up manuscript version for detail.

4- Concerning Alnus, don't you think that Alnus takes the place of Quercus and Lithocarpus/Castanopsis in the upper 4kyr?
**Response:** Yes, *Alnus* pollen percentages took the place of evergreen oaks and *Lithocarpus/Castanopsis* pollen percentages in the upper 4 kyr. However, due to high representation of *Alnus* pollen (Modern pollen rain results in southwestern China show that *Alnus* pollen percentages are more than 70% in a patch of *Alnus* forest), *Alnus* forest around Qinghai Lake during the period 4.3-0.8 ka may be relatively low density or it may be not pure *Alnus* forest. Thus, we added "This deduction is verified by high representation of *Alnus* pollen" in line 22 page 18 of the marked-up manuscript version.

5- Page 19 - Line 26 to 30 I don't understand the first sentence?? "dominant species are not evident"??
The other two sentences need a following one to explain what implies the first two. When reading this paragraph, at the end we wonder its aim.

**Response: "**dominant species are not evident" means "there are no obvious dominant species". We changed **"**dominant species are not evident" into "there are no obvious dominant species" in ==line 2 page 20== of the marked-up manuscript version.

We added a sentence "These indicate that an evident ameliorating climate was more favorable to plant diversity than an optimum climate." at the end of this paragraph.

6- Page 20 the sentence "the low fire……. declining" is not so clear. You talk about "the decrease in richness and increase in evenness" I think that talk about the vegetation and then mark "vegetation richness" and "vegetation evenness"

And justify the last sentence.

**Response:** We marked "vegetation richness" and "vegetation evenness" in ==line 11 page 20== of the marked-up manuscript version. About the last sentence "Overall, human activity is likely to be the main driver of altered fire regimes after 4.3 ka and the response of plant diversity (richness and evenness) during this period", we consider that the 5.4 section justified that human activity is likely to be the main driver of altered fire regimes after 4.3 ka and this paragraph explained that human activity is likely to be the main driver of altered the response of plant diversity (richness and evenness) to fire activity during this period.

7- Last paragraph of the conclusion: delete" it is necessary that…. " and begin with "further lacustrine sites…… China" and continue with "remain necessary to improve our understanding …interactions"

**Response:** Done.

8- Acknowledgments: please do not forget to thank the reviewers even anonymous.

**Response:** We added "The paper has been strongly improved thanks to the comments of Editor Nathalie Combourieu-Nebout and four anonymous reviewers." in the Acknowledgments section.

9- References: please verify the bibliography. I did not find Birks and Line, 1992; Li, 2004 in the text

**Response:** We verified the bibliography again. We changed "Birk and Line" in ==line 10 page 10== into "Birks and Line". "Li, 2004" is in ==line 11 page16== in the origin manuscript.

Suggestions for modifications in the text:

Line 9 page 13 add "that" between reveals and fire

**Response:** Done.

Line 17 page 14 "HS1 occurs" or "is dated between"

**Response:** Done.

Line 23 page 14 "this climatic tendency…" can be replaced by "this climatic trend is also recorded in other data from"

**Response:** Done.

Line 8 to 10 page 15 rephrase this sentence, it is not clear or cut the two ideas in two sentences

**Response:** We rephrase this sentence into "One reason may be that the relatively low temperature was not favourable to fuel desiccation. However, the more important reason may be that the climate was not too dry or rainfall seasonality was relatively low in the study area, because drought is the key factor influencing fire occurrences in southwestern China (Gu et al., 2008)." Please see ==lines 10-13 page 15== in the marked-up manuscript version for detail.

Line 22 page 15 change "in the dry period" by "during the dry period"

**Response:** Done.

Line 23 page 15 same remarks concerning the wet periods.
**Response:** Done.

Line 24 page 15 what is the HE profile??
**Response:** The HE profile is a well-dated peat profile with a thickness of 148 cm, located in the northern part of the Sanjiang Plain in the Heilongjiang Province. We changed this sentence into "A well-dated peat profile (i.e. the HE profile) with high resolution charcoal and pollen records from the Sanjiang Plain in northeastern China …" in the revised manuscript. Please see lines 26-27 page 15 in the marked-up manuscript version for detail.

Line 24 page 15 add "in" before "northwestern China"
**Response:** Done.

Line 19 to 23 page 17 rephrase. It is not clear. Simplify or cut in two sentences.
**Response:** Done. Please see lines 22-26 page 17 in the marked-up manuscript version for detail.

Line 21 page 18 delete "the" before "other regions"
**Response:** Done.

Line 16 page 19 change "for" by "of" before "fire-sensitive"
**Response:** Done.

Line 20 page 19 "that floristic… " I do not understand what you mean. I prefer "where floristic…" but is it what you want to say??
**Response:** Yes, it is. We changed "that" into "where".

Please pay attention to these remarks and submit a new revised version and I will state about the acceptation of your manuscript.

Best regards
Nathalie Combourieu-Nebout
**Response:** We appreciate very much your comments on the manuscript. According to your comments and remarks, we revised carefully the manuscript and address each comment with explanation as above. Please see the file "Response to the Editor" and the marked-up manuscript version for detail.
    We hope we successfully complied with the comments and remarks, and we appreciate your detailed reviews and suggestions.

[revised manuscript text omitted]

